# Neuromedin U signaling regulates retrieval of learned salt avoidance in a *C. elegans* gustatory circuit

Jan Watteyne [1], Katleen Peymen[1], Petrus Van der Auwera [1], Charline Borghgraef [1], Elke Vandewyer[1], Sara Van Damme[1], Iene Rutten[2], Jeroen Lammertyn [2], Rob Jelier[3], Liliane Schoofs[1✉] & Isabel Beets [1✉]

Learning and memory are regulated by neuromodulatory pathways, but the contribution and temporal requirement of most neuromodulators in a learning circuit are unknown. Here we identify the evolutionarily conserved neuromedin U (NMU) neuropeptide family as a regulator of *C. elegans* gustatory aversive learning. The NMU homolog CAPA-1 and its receptor NMUR-1 are required for the retrieval of learned salt avoidance. Gustatory aversive learning requires the release of CAPA-1 neuropeptides from sensory ASG neurons that respond to salt stimuli in an experience-dependent manner. Optogenetic silencing of CAPA-1 neurons blocks the expression, but not the acquisition, of learned salt avoidance. CAPA-1 signals through NMUR-1 in AFD sensory neurons to modulate two navigational strategies for salt chemotaxis. Aversive conditioning thus recruits NMU signaling to modulate locomotor programs for expressing learned avoidance behavior. Because NMU signaling is conserved across bilaterian animals, our findings incite further research into its function in other learning circuits.

[1] Department of Biology, KU Leuven, 3000 Leuven, Belgium. [2] Department of Biosystems, KU Leuven, 3000 Leuven, Belgium. [3] Department of Microbial and Molecular Systems, KU Leuven, 3000 Leuven, Belgium. ✉email: liliane.schoofs@kuleuven.be; isabel.beets@kuleuven.be

Aversive learning is crucial for animals to adapt behavior, which increases their chance of survival. Learning triggers experience-driven changes in neuronal signaling and many of the underlying molecular mechanisms are evolutionarily conserved[1]. One main class of molecules involved in plasticity are neuromodulators, which can either shift global brain activity or have more focused actions by altering the properties of individual synapses, neurons, or local circuits[2,3]. Neuromodulators mainly function by activating G protein-coupled receptors (GPCRs)[3]. Strong evidence across metazoan models implicates neuromodulatory systems in learning and memory[4,5], but for most mechanisms underlying these effects remain elusive.

While monoamines have received considerable attention for their role in cognition[6], neuropeptide signaling represents another route of neuromodulation. Neuropeptides are an ancient and diverse class of secreted messengers that have recently been recognized as modulators of learning circuits[5,7,8]. This has fueled research into peptidergic systems as potential therapeutic targets for cognitive disorders[9,10]. While some neuropeptides have been implicated in experience-dependent plasticity, uncovering their actions in learning circuits has remained a challenge, in part because many neuropeptides signal extrasynaptically and interact with other neuromodulatory pathways[11,12]. Much of our current understanding is derived from exogenous application of neuropeptides, but this approach has limited spatial and temporal resolution to characterize peptidergic signaling in vivo[5,10]. Furthermore, the sheer number of conserved neuropeptides and receptors expressed in brain regions involved in memory[5,10,13] suggests that additional neuropeptides are yet to be discovered as neuromodulators of learning circuits that likely play an evolutionarily conserved role in experience-dependent plasticity.

Neuromedin U (NMU) represents an ancient family of neuropeptides with members in most bilaterian animals[7]. NMU is known for its profound effects on feeding and locomotor activity[14–18]. However, several lines of evidence suggest that NMU neuropeptides are also involved in memory and may be ancient modulators of learning circuits. NMU peptides and their receptors are widely expressed in memory centers including the vertebrate hippocampus and the insect mushroom bodies[13–15,19]. Central administration of NMU peptides alleviates inflammation-induced amnesia[20,21] and also affects reward-related contextual learning[22]. Thus, NMU peptides may be neuromodulators of learning circuits, although the underlying mechanisms remain poorly defined.

The nematode C. elegans, with its compact and mapped nervous system[23,24], provides a powerful model for dissecting learning and neuropeptidergic circuits. We previously identified an NMU-related neuropeptide precursor in C. elegans, named CAPA-1/NLP-44, which is orthologous to vertebrate NMU and its related capability (capa) derived neuropeptides in Drosophila[25]. The C. elegans genome also encodes three orthologs of the NMU receptor (NMUR) family, called NMUR-1 to NMUR-3[7]. These receptors, in particular nmur-1, are expressed in sensory neurons that mediate the transduction of gustatory among other cues[26], and in interneurons of which AIA was identified as integration center for aversive learning[27,28]. Based on these findings and the proposed role of NMU in vertebrate learning, we hypothesized that CAPA-1 signaling regulates gustatory aversive learning in C. elegans.

Behavioral adaptations and neural mechanisms underlying C. elegans gustatory aversive learning are well characterized[29–33], providing a valuable framework to dissect the precise effects of neuropeptidergic modulators. We previously found that this type of associative learning is regulated by nematocin, a neuropeptide of the ancient vasopressin/oxytocin family[34]. A broad array of neuropeptides operates within the C. elegans nervous

system, many of which are evolutionarily conserved between worms and humans but remain functionally uncharacterized[7,35]. Furthermore, our understanding on how individual modulators contribute to a learned behavior in a defined circuit is limited. Behaviorally, it is unclear whether neuropeptides regulate specific aspects of a learned behavior or rather serve as general modulators that coordinately control distinct conditioned responses to a stimulus. The temporal requirement of peptidergic neurons in learning circuits also remains largely unexplored.

Here, we address these questions by neurogenetic manipulations of the NMU system in the C. elegans circuit for gustatory aversive learning. Our work indicates that NMU signaling modulates learning, and shows that NMU neurons are specifically required for the retrieval of learned aversion. NMU neuropeptides released from ASG sensory neurons signal via their receptor NMUR-1 on AFD neurons to coordinately regulate different behavioral strategies underlying gustatory aversive learning. This suggests that NMU might mediate similar functions across learning circuits given the broad evolutionary conservation of this signaling system.

## Results

**Mutants for nmur-1 show impaired gustatory aversive learning**. C. elegans has three predicted NMU receptor orthologs, named NMUR-1 to NMUR-3[7] (Fig. 1a). To investigate whether this receptor family is involved in gustatory aversive learning, we analyzed the performance of nmur loss-of-function mutants (Fig. 1b) in an established associative learning paradigm[32,34]. Similar to classical aversive conditioning, we trained worms by pairing a conditioned salt stimulus with the absence of food during a 15-min training period (Fig. 1c). We then tested salt chemotaxis behavior by putting worms in the center of a quadrant plate and calculated a chemotaxis index based on the number of animals that migrated towards or away from NaCl after 10 min (Fig. 1c). This time window allows testing for defects in learned salt aversion before animals revert to NaCl attraction[36]. Mock-conditioned worms not pre-exposed to salt were strongly attracted to NaCl and wild-type animals learned to avoid salt when they previously experienced it in the absence of food (Fig. 1d). Consistent with our hypothesis, nmur-1 mutants were defective in learning and still attracted to salt after NaCl-conditioning (Fig. 1d). Mutants for nmur-2 and nmur-3 had no learning defects (Fig. 1d), suggesting a role specific for nmur-1 in gustatory aversive learning.

The defective learning phenotype of nmur-1 animals could emanate from a failure to associate salt with the lack of food or may result from general defects in salt chemotaxis behavior or osmotic homeostasis. To test this, we quantified chemotaxis behavior of nmur-1 animals in response to different stimuli and training regimes in the quadrant assay (Supplementary Fig. 1). Chemotaxis behavior towards increasing NaCl concentrations was unaffected in nmur-1 animals (Supplementary Fig. 1a), indicating that salt sensing is not affected in nmur-1 mutants. To determine if the learning defect of nmur-1 results from a defect in osmotic homeostasis, we replaced NaCl with glycerol at the same osmolarity in our conditioning protocol (Supplementary Fig. 1b). Pairing glycerol with food absence did not decrease NaCl attraction in wild-type and nmur-1 animals (Supplementary Fig. 1b), suggesting that salt chemotaxis is not differentially affected by exposure to increased osmolarity in nmur-1 mutants. Conversely, we questioned whether nmur-1 animals differentially respond to increased osmolarity after conditioning with NaCl. For this, we replaced NaCl in chemotaxis assay plates with glycerol at the same osmolarity (Supplementary Fig. 1c). Wild-

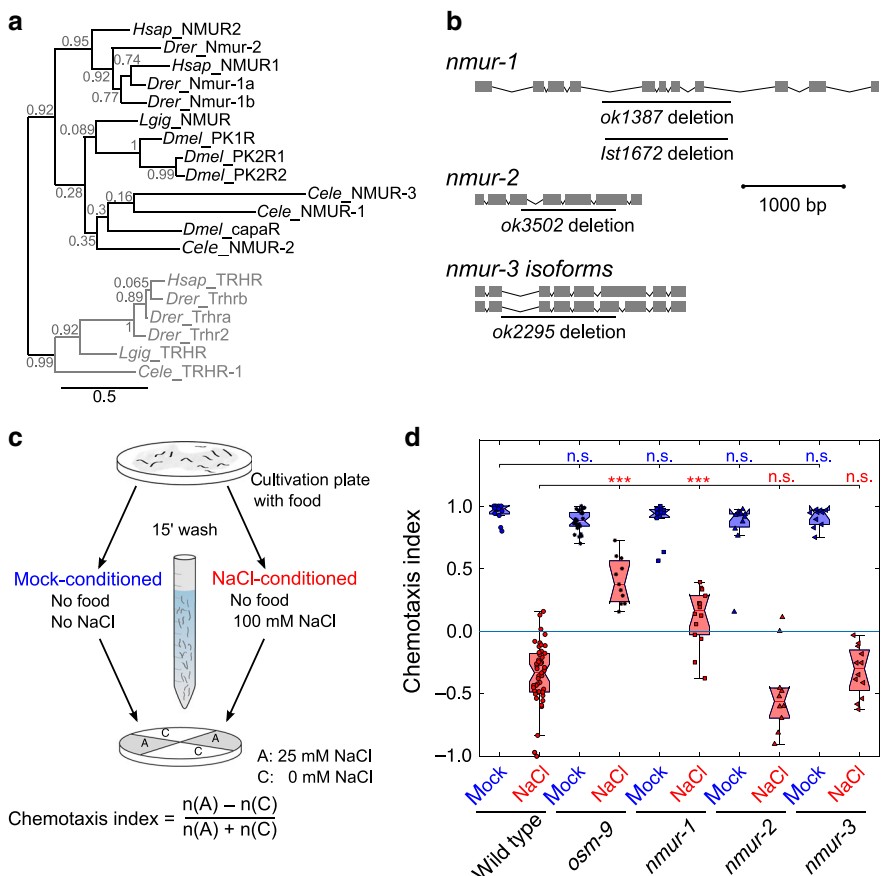

**Fig. 1 The *C. elegans* neuromedin U receptor ortholog *nmur-1* regulates gustatory aversive learning. a** Maximum likelihood tree of the phylogenetic relationship between neuromedin U receptor (NMUR) orthologs of *C. elegans* (*Cele*), *D. melanogaster* (*Dmel*), *L. gigantea* (*Lgig*), *D. rerio* (*Drer*) and human (*Hsap*). Thyrotropin-releasing hormone receptors (TRHRs) are used as outgroup to root the tree, because this receptor family was shown to be most related to bilaterian NMURs[7]. Branch node labels correspond to likelihood ratio test values. **b** Gene organization of three predicted *nmur* orthologs in *C. elegans*, named *nmur-1* to *nmur-3*. Boxes represent exons and lines show intronic sequences. Bars indicate the position of deletions in loss-of-function alleles used in this study. **c** Overview of quadrant assay for gustatory aversive learning. The intrinsic attraction to NaCl is reversed by conditioning worms with salt in the absence of food during a short-term (15 min) training period. Mock-conditioned animals, which have not been pre-exposed to salt, are used as control. After conditioning, NaCl chemotaxis behavior is measured by determining the distribution of worms on quadrant plates of which two opposing quadrants are supplemented with NaCl and a chemotaxis index is calculated. **d** Gustatory aversive learning of mutants defective in NMUR signaling: *nmur-1* (*ok1387*), *nmur-2* (*ok3502*) and *nmur-3* (*ok2295*). A mutant of *osm-9*, a TRPV channel subunit required for learned salt aversion[34], is used as positive control. Significances determined by two-way ANOVA and Tukey's post hoc test. n.s. not significant; ***$p \leq 0.001$. From left to right $n = 13, 47, 26, 11, 12, 14, 11, 10, 10, 12$ experimental repeats. In this and all subsequent figures, data from mock-conditioned worms is depicted in blue, while red markers denote NaCl-conditioned worms. The central mark in boxplots on this and all other figures indicates the median; the bottom and top edges reflect the 25th and 75th percentiles, respectively. Whiskers extend to the most extreme data points not considered as an outlier. Outliers are always included in statistical analyses.

type animals showed mild attraction to glycerol and this response was reduced in NaCl-conditioned animals. However, the difference in behavioral switch after NaCl conditioning was much smaller when assaying chemotaxis to glycerol than to NaCl, which suggests the latter is a learned response rather than an effect of osmotic acclimation. Mutants of *nmur-1* behaved like wild-type animals in glycerol chemotaxis assays (Supplementary Fig. 1c), indicating that *nmur-1* animals are defective in gustatory aversive learning rather than salt sensing or the ability to maintain osmotic balance.

If *nmur-1* mutants are defective in associating salt with absence of food, then animals should display normal salt attraction when salt is paired with food. To test this, we performed conditioning on agar plates to which food can be added, instead of pre-exposing animals to NaCl in liquid (Supplementary Fig. 1d). We conditioned animals for 30 min according to the reported conditioning period required for learned salt aversion when conditioning on plates[32]. As expected, *nmur-1* mutants showed

normal salt chemotaxis behavior when salt was paired with food but had learning defects when trained with salt in the absence of food (Supplementary Fig. 1d).

We then asked if this learning defect could be extended to other aversive experiences and tested *nmur-1* animals after pairing salt with an aversive concentration of the odorant benzaldehyde[32]. We conditioned animals with undiluted benzaldehyde for 30 min on plates with food, in the absence or presence of 100 mM NaCl (Supplementary Fig. 1e). Mock-conditioned worms, only pre-exposed to benzaldehyde, were strongly attracted to NaCl and wild-type animals learned to avoid salt when they previously experienced it in the presence of benzaldehyde (Supplementary Fig. 1e). By contrast, *nmur-1* mutants were still attracted to NaCl after exposure to salt with benzaldehyde (Supplementary Fig. 1e). Thus, *nmur-1* mutants are impaired in learning associations between salt and different aversive experiences. Taken together, these findings suggest that *nmur-1* promotes gustatory aversive learning.

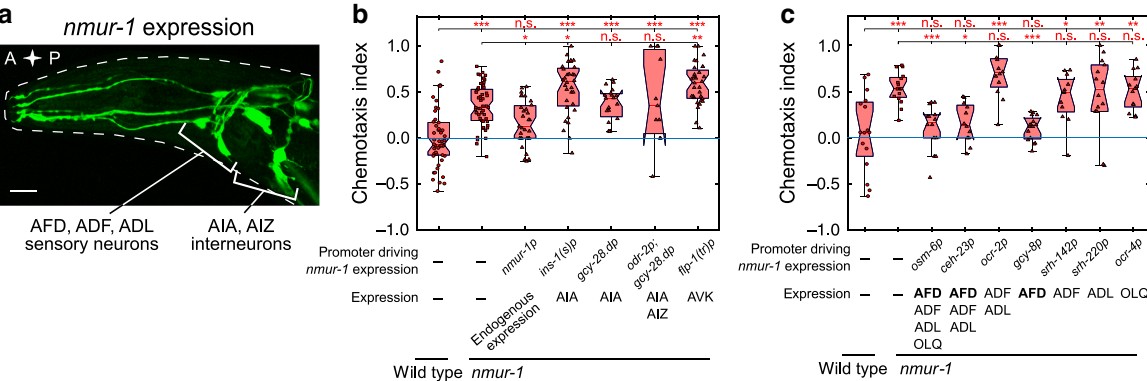

**Fig. 2 nmur-1 is required in AFD sensory neurons for gustatory aversive learning. a** Transgenic adult hermaphrodite *C. elegans* expressing a bicistronic GFP construct including the promoter, cDNA and 3'UTR of the *nmur-1* gene (*nmur-1p::nmur-1::SL2::gfp*). Scale bar, 35 μm. **b, c** Chemotaxis index of NaCl-conditioned *nmur-1 (ok1387)* animals in which *nmur-1* expression is restored under the control of its **b** endogenous promoter or in selected interneurons and **c** in sensory neurons. Significances determined by one-way ANOVA and Tukey's post hoc test. n.s. not significant; *$p \leq 0.05$; **$p \leq 0.01$; ***$p \leq 0.001$. **b** From left to right $n = 50, 50, 31, 32, 19, 12, 32$ experimental repeats. **c** From left to right $n = 16, 16, 12, 14, 12, 13, 12, 14, 16$ experimental repeats. Boxplots show medians, 25th and 75th percentiles as box limits. The whiskers extend to the most extreme data points not considered as outlier. .

**NMUR-1 is required in AFD for gustatory aversive learning**. A fluorescent reporter transgene (Fig. 2a) recapitulated the reported expression pattern for *nmur-1* in approximately 15 sensory neurons (including ADF, ADL and AFD) and interneurons (AIA, AIZ, AVK, among others)[28]. To confirm the role of *nmur-1* in gustatory aversive learning, we restored *nmur-1* expression under its endogenous promoter in the mutant and tested salt chemotaxis in the quadrant assay. This fully rescued learned salt aversion of *nmur-1* animals (Fig. 2b).

Next, we asked in which cell(s) *nmur-1* is required for gustatory aversive learning (Fig. 2b-c). Expressing wild-type copies of *nmur-1* in AFD sensory neurons using various promoters, including the AFD-specific *gcy-8* promoter[37], fully restored learned salt aversion (Fig. 2c). Expression of *nmur-1* in other sensory neurons and interneurons did not rescue the mutant phenotype (Fig. 2b-c). These results indicate that NMUR-1 is required in AFD for gustatory aversive learning.

**NMUR-1 adapts behavioral strategies in learned salt aversion**. Behavioral tracking on NaCl gradients previously showed that learned salt aversion requires the experience-dependent modulation of two main navigational strategies. Untrained worms increase the duration of forward movement (runs) when salt concentrations increase locally, called biased random walk[38], and they actively steer towards attractive NaCl concentrations, known as klinotaxis[29]. After aversive conditioning with salt and food absence, both strategies are reversed to mediate NaCl avoidance instead of salt attraction[29,33,39]. As specific neuronal substrates have been delineated for each strategy[29,33,39], we wondered if *nmur-1* is specifically involved in one of these responses, or if it acts as a general regulator of learned salt avoidance modulating both behavioral strategies. Therefore, we tested gustatory aversive learning of *nmur-1* mutants on linear NaCl gradients, which allows more detailed analysis of the locomotory patterns underlying salt chemotaxis behavior.

For this, we set up a tracking platform to monitor trajectories of individual worms and quantify locomotion parameters relative to a linear 0–100 mM NaCl gradient (Fig. 3a and Supplementary Fig. 2a, b). We first examined how gustatory aversive learning in wild-type animals is manifested on these gradients by measuring the position of worms during a 30-min time window immediately after training. Mock-conditioned wild-type animals were attracted to salt and had migrated up the gradient after ten minutes. NaCl-conditioned animals avoided high salt

concentrations by migrating to lower positions on the gradient in this period, and then gradually moved back to higher salt concentrations (Fig. 3b and Supplementary Fig. 2c). Besides positions on the gradient, we calculated a chemotactic index reflecting chemotactic rate and direction[39] during the 10-min time window after release on the NaCl gradient (Fig. 3c), corresponding to the time interval in which worms showed the largest chemotactic activity (Supplementary Fig. 2g). As expected, mock-conditioned wild-type animals had a positive chemotactic index, whereas NaCl-conditioned animals showed negative values due to salt avoidance (Fig. 3c). We also calculated indices to describe different behavioral strategies for salt chemotaxis, i.e. biased random walk and klinotaxis[39]. In line with former studies[29,39], both navigational strategies were modulated by aversive experience, because biased random walk and klinotaxis indices of wild-type animals reversed after NaCl-conditioning (Fig. 3d, e).

We next examined gustatory aversive learning of *nmur-1* mutants on NaCl gradients. Similar to the quadrant assay (Fig. 1d), mock-conditioned *nmur-1* animals displayed normal salt chemotaxis behavior, while NaCl-conditioned mutants showed less aversion for high NaCl concentrations than wild type (Fig. 3b-c). We observed significant differences in both biased random walk and klinotaxis of wild type and *nmur-1* mutants after aversive conditioning (Fig. 3d-e). To strengthen the behavioral phenotypes of *nmur-1*, we generated a second *nmur-1* mutant allele, *lst1672*, using CRISPR (Fig. 1b and Methods). This mutant was also defective in adapting biased random walk and klinotaxis after NaCl conditioning (Fig. 3b–e). These results suggest that NMUR-1 is involved in the experience-dependent modulation of both the length of runs and the reorientation on a NaCl gradient.

We then asked whether the behavioral phenotypes of *nmur-1* mutants depend on the conditioning time, because a mutation in *nmur-1* may affect the training period required for learning salt aversion. We conditioned animals for 7.5, 15, 30, and 60 min (Supplementary Fig. 2d–g). Wild-type animals showed stronger salt aversion with increased conditioning times (Supplementary Fig. 2f, g). Mutants of *nmur-1* were impaired in gustatory aversive learning for each conditioning time tested (Supplementary Fig. 2d, e), suggesting that the learning defect does not result from alterations in the conditioning period required for learning. We also examined *nmur-1* phenotypes on different linear NaCl gradients ranging from 0 to 50, 100 or 150 mM NaCl

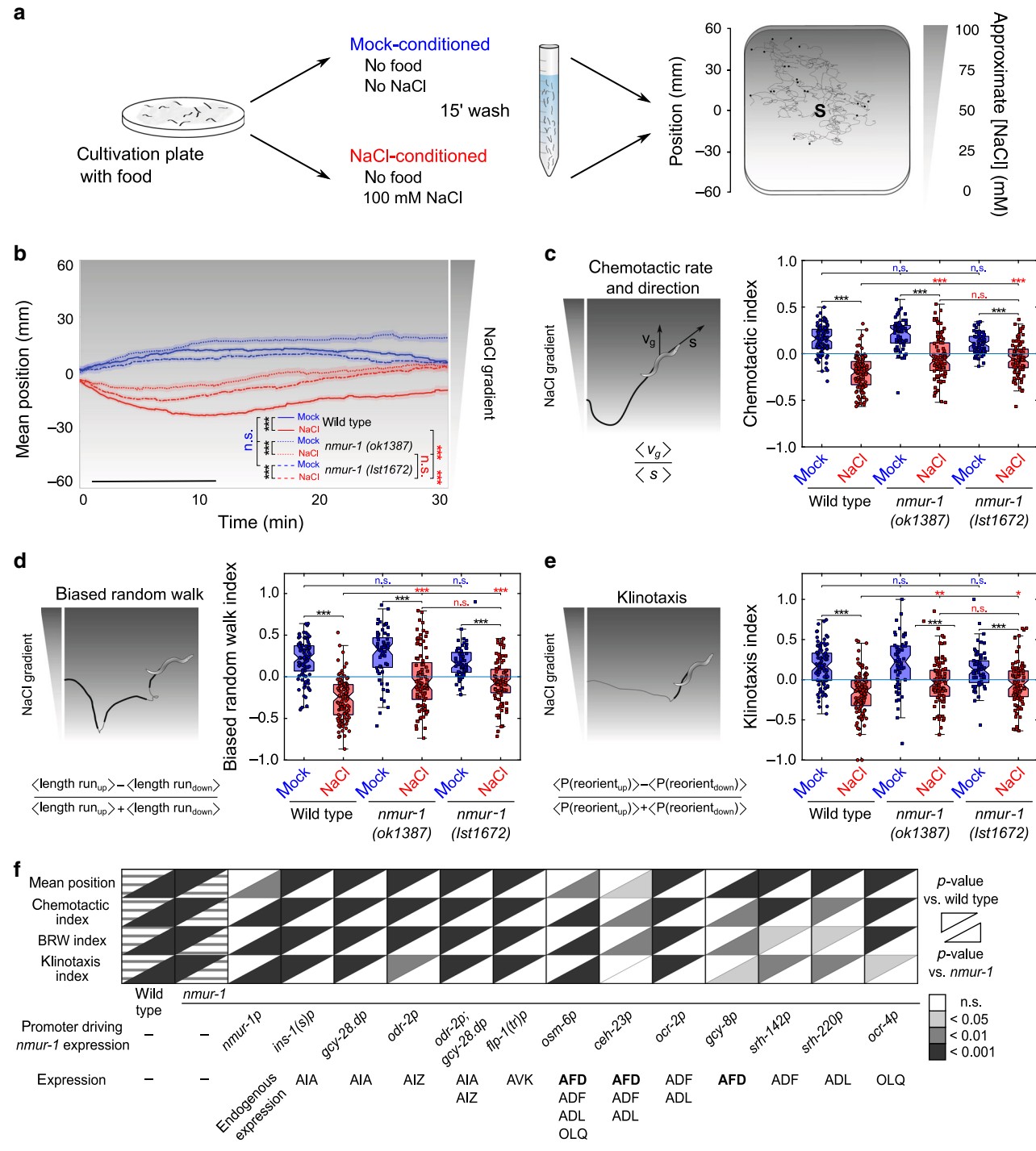

(Supplementary Fig. 2b). *nmur-1* animals were defective in learning salt aversion on each gradient tested (Supplementary Fig. 3a–c).

In accordance with results obtained in the quadrant assay (Fig. 2b, c), *nmur-1* phenotypes on NaCl gradients were rescued by reintroducing wild-type *nmur-1* under selective promoters driving expression in AFD, but not by expressing *nmur-1* in other sensory neurons or interneurons (Fig. 3f and Supplementary Figs. 4 and 5). Thus, the mutant phenotypes and cellular focus of *nmur-1* are consistent across different salt chemotaxis assays. To examine further if AFD is involved in salt chemotaxis behavior, we tested AFD-ablated animals expressing cell-death-inducing caspases under the AFD-specific *gcy-8* promoter[40,41]. These

animals were defective in NaCl chemotaxis and showed higher immobility on NaCl gradients than wild type (Supplementary Fig. 6a–c). Taken together, our data show that AFD signaling contributes to NaCl chemotaxis behavior and that *nmur-1* is required in these neurons for gustatory aversive learning.

**CAPA-1/NLP-44 NMU-like neuropeptides are ligands of NMUR-1.** Next, we set out to identify the ligand(s) of the orphan receptor NMUR-1 by reverse pharmacology. We expressed NMUR-1 in Chinese Hamster Ovary (CHO) cells that were challenged with a synthetic library of over 300 *C. elegans* peptides, and monitored GPCR activation by a calcium reporter assay

**Fig. 3 NMUR-1 signaling in AFD modulates distinct behavioral strategies underlying learned salt avoidance. a** Tracking setup for measuring gustatory aversive learning on NaCl gradients. NaCl chemotaxis behavior is monitored on 0 to 100 mM linear NaCl gradients. Conditioned worms are released from the gradient's center (S, source) and allowed to navigate for 30 min. Tracking software extracts positional data and determines behavioral indices for forward movement and turning behavior relative to the gradient. **b** Average positions of wild type, *nmur-1 (ok1387)* and *nmur-1 (lst1672)* mutants on the linear NaCl gradient through time after mock- and NaCl-conditioning. In this and all subsequent figures, black bars indicate the time interval used for statistical comparison of the mean position on the gradient, and for calculating chemotactic, biased random walk and klinotaxis indices. Shaded regions represent S.E.M. **c** The chemotactic index quantifies the preferential migration vector of each individual and is calculated as the ratio between the mean velocity of an individual trajectory along the gradient direction $\langle v_g \rangle$ and the mean crawling speed $\langle s \rangle$. **d** Quantification of biased random walk behavior, assessing the relative duration of forward runs when navigating up the salt gradient per worm. **e** The klinotaxis index reflects the behavioral strategy by which worms actively steer towards attractive NaCl concentrations, quantified as the fractional difference in the relative probabilities of turns reorienting individual animals up or down the gradient. **b–e** From left to right $n = 92, 110, 64, 92, 66, 87$ animals per condition. **f** Summary statistics of rescue experiments for *nmur-1* learned salt aversion defect. For each of four behavioral parameters, significances are relative to NaCl-conditioned wild-type and *nmur-1 (ok1387)* animals. Raw data on Supplementary Figs. 4 and 5. Statistical analysis by two-way ANOVA and Tukey's HSD post hoc test in **b–e**, and one-way ANOVA with Tukey's post hoc test or two-sided Kruskal–Wallis with Dunn's post hoc test if one of the conditions was not normally distributed in **f**. n.s. not significant; *$p \leq 0.05$; **$p \leq 0.01$; ***$p \leq 0.001$. Boxplots show medians, 25th and 75th percentiles as box limits. Whiskers extend to the most extreme data points not considered as outlier.

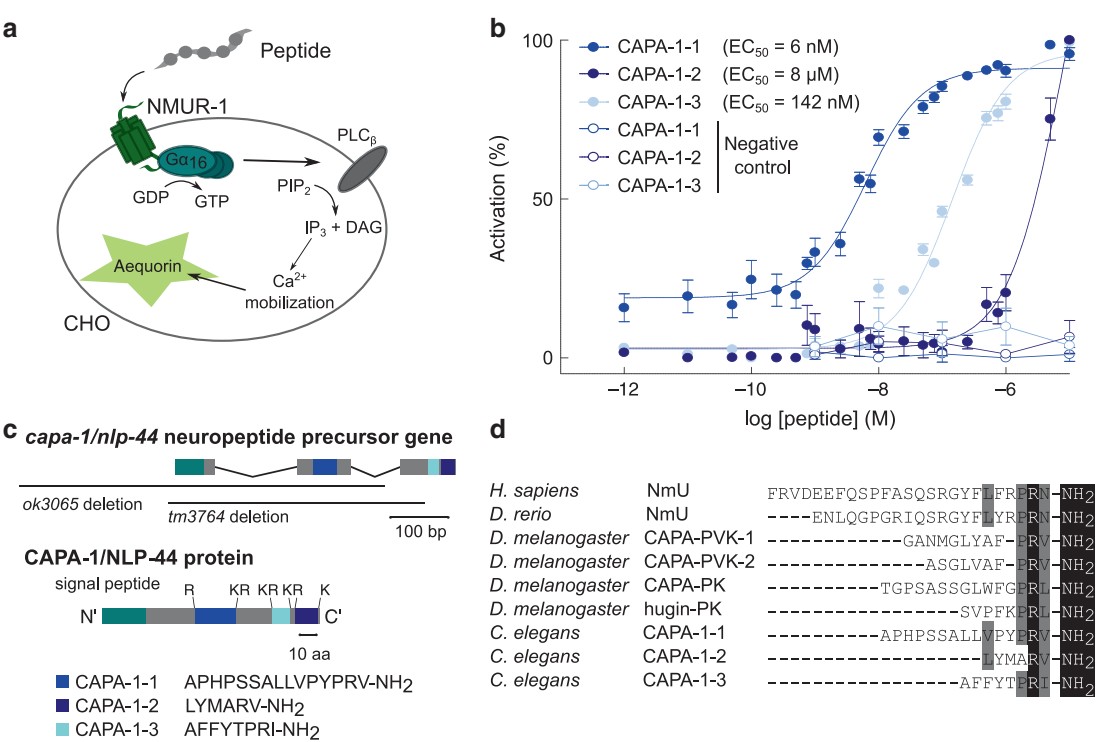

**Fig. 4 CAPA-1 NMU-like neuropeptides activate NMUR-1 in a cellular assay for GPCR activation. a** Luminescence-based calcium mobilization assay for measuring GPCR activation. NMUR-1 is expressed in CHO cells that stably co-express the promiscuous human $G\alpha_{16}$ subunit, which couples receptor activation to calcium release from intracellular storage sites. Intracellular calcium levels are monitored using the calcium indicator aequorin. **b** Calcium responses of CHO cells expressing NMUR-1 are shown relative (%) to the highest value (100% activation) after normalization to the total calcium response. Cells transfected with an empty pcDNA3.1 vector are used as a control. Error bars show S.E.M. CAPA-1-1 $n = 6$; CAPA-1-2 $n = 6$; CAPA-1-3 $n = 5$ experimental repeats. **c** The *capa-1/nlp-44* gene encodes a neuropeptide precursor that harbors three predicted peptide sequences (CAPA-1-1, -2, and -3), which are flanked by mono- and dibasic cleaving sites. Black bars indicate the positions of deletions in *capa-1/nlp-44* mutant alleles used in this study. **d** Sequence comparison of *C. elegans* CAPA-1 neuropeptides and NmU neuropeptides from *H. sapiens*, *D. rerio*, and *D. melanogaster*. Conserved C-terminal features are highlighted in black. Residues with similar physicochemical properties are colored gray.

(Fig. 4a). Only neuropeptides encoded by the *capa-1/nlp-44* gene elicited a calcium response in NMUR-1-expressing cells. Cells transfected with an empty vector showed no response (Fig. 4b).

*C. elegans* NLP-44 is a neuropeptide precursor of the NMU family[7,25]. It encodes three predicted peptides with a C-terminal YXPR(I/V)-NH2 motif, resembling the PRX-NH2 motifs of *Drosophila* and mammalian NMU peptides[42] (Fig. 4c, d). In *Drosophila* NMU-like peptides are encoded by the *capability* and *hugin* genes. Due to the sequence similarity of *nlp-44* with the insect *capability* gene, *C. elegans nlp-44* has been named *capa-1*[25].

We determined the potency of individual CAPA-1 synthetic peptides to activate NMUR-1. CAPA-1-1 activated NMUR-1 with a half maximal effective concentration ($EC_{50}$) of 6 nM, while CAPA-1-2 and CAPA-1-3 showed higher $EC_{50}$ values of 8 μM and 142 nM, respectively (Fig. 4b).

**CAPA-1 is required in ASG for gustatory aversive learning.** Because CAPA-1 peptides activated NMUR-1 in vitro, we hypothesized that these neuropeptides promote learned salt aversion through NMUR-1 signaling. We first tested gustatory

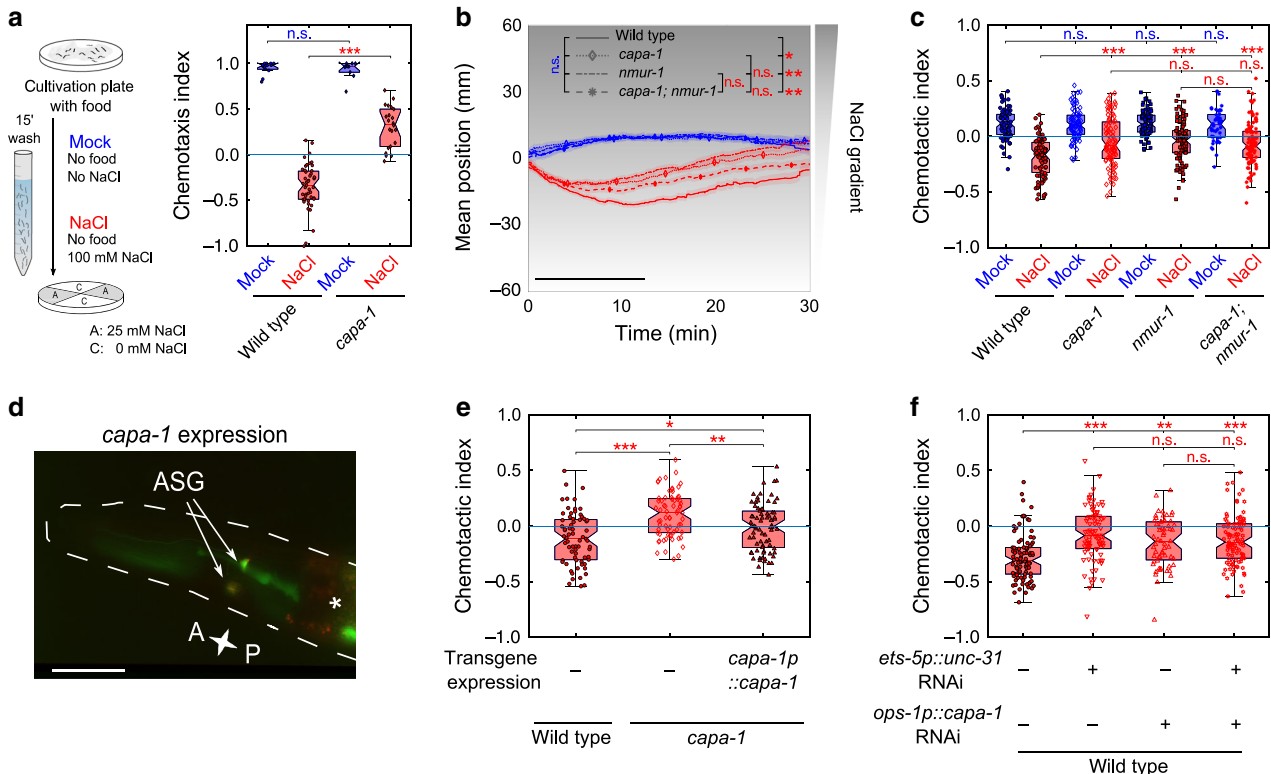

**Fig. 5 CAPA-1 NMU-like neuropeptides signal from ASG neurons to promote gustatory aversive learning through NMUR-1. a** Gustatory aversive learning behavior of *capa-1 (ok3065)* mutant animals in the quadrant chemotaxis assay. From left to right $n = 13, 47, 10, 22$ experimental repeats. **b** Mean position on a 0–100 mM NaCl gradient through time and **c** the corresponding chemotactic indices of *capa-1 (ok3065)*, *nmur-1 (lst1672)* and *capa-1 (ok3065); nmur-1 (lst1672)* double mutants after mock- and NaCl-conditioning. Black bar indicates the time interval for calculating the mean gradient position and the chemotactic index, but also the biased random walk and klinotaxis indices shown in Supplementary Fig. 7b. From left to right $n = 79, 89, 79, 98, 83, 100,$ 48, 101 animals per genotype. **d** Co-localization of a bicistronic GFP construct harboring the promoter, genomic DNA and 3′UTR of the *capa-1* gene (*capa-1p::capa-1::SL2::gfp::3′UTRcapa-1*) with ASG-specific mCherry expression (*gcy-15p::mCherry*[43]). The asterisk marks fluorescence in the intestine from the *elt-2p::gfp* co-injection marker. Scale bar, 35 µm. **e** Chemotactic behavior of NaCl-conditioned *capa-1 (ok3065)* animals in which wild-type copies of the *capa-1* gene are transgenically expressed under the control of its promoter sequence (*capa-1p::capa-1::SL2::gfp*). The corresponding mean population position on the gradient, biased random walk and klinotaxis indices are shown on Supplementary Fig. 7d. From left to right $n = 94, 83, 88$ animals per genotype. **f** Chemotactic behavior of NaCl-conditioned animals with RNAi knockdown of *unc-31* using the *ets-5* promoter (expressed in ASG and BAG)[47], RNAi knockdown of *capa-1* using the ASG-specific *ops-1* promoter[48], or both. Corresponding mean population position on the gradient, biased random walk and klinotaxis indices shown on Supplementary Fig. 7e. From left to right $n = 92, 88, 60, 102$ animals per genotype. Statistical analysis by two-way ANOVA followed by Tukey's HSD post hoc test in **a–c**, one-way ANOVA and Tukey's post hoc test (**e**) and two-sided Kruskal–Wallis with Dunn's post hoc test (**f**). n.s. not significant; *$p \leq 0.05$; **$p \leq 0.01$; ***$p \leq 0.001$. Boxplots show medians, 25th and 75th percentiles as box limits. The whiskers extend to the most extreme data points not considered as outlier.

aversive learning of *capa-1* mutants (Fig. 4c) in the quadrant assay (Fig. 5a). As expected, *capa-1* animals were still attracted to NaCl after experiencing it in absence of food (Fig. 5a). This learning defect did not result from general defects in salt chemotaxis behavior or in responses to osmotic stimuli (Supplementary Fig. 1a–c). Furthermore, *capa-1* animals showed normal salt chemotaxis behavior when NaCl was paired with food but were defective in associating salt with different aversive experiences (Supplementary Fig. 1d, e). These results suggest that CAPA-1 neuropeptides are required for gustatory aversive learning.

Next, we determined if loss of *capa-1* disrupts locomotory patterns for gustatory aversive learning on NaCl gradients. After NaCl conditioning, *capa-1* mutants showed significant differences in biased random walk and klinotaxis compared to wild type (Supplementary Fig. 7a). An independent *capa-1* mutant allele displayed similar behavioral phenotypes (Supplementary Fig. 7a). To determine if CAPA-1 peptides signal through NMUR-1 in vivo, we tested gustatory aversive learning in a *capa-1; nmur-1* double mutant. Mutants lacking both *capa-1* and *nmur-1* did not

show an additive defect compared to single mutants (Fig. 5b-c and Supplementary Fig. 7b). Thus, *capa-1* functions in the same genetic pathway as *nmur-1*. Worms overexpressing *capa-1* behaved like wild-type animals, suggesting that salt chemotaxis behavior is not affected by increased *capa-1* expression (Supplementary Fig. 7c).

Using a bicistronic GFP reporter transgene, we found that *capa-1* is expressed in a single pair of chemosensory neurons (Fig. 5d), in accordance with immunostainings using a CAPA-1 antiserum[25]. Co-localization of GFP with mCherry, expressed from the ASG-specific *gcy-15* promoter[43], identified these cells as ASG (Fig. 5d). ASG is involved in detecting food and water-soluble attractants, such as NaCl[26,44,45]. Restoring *capa-1* expression in ASG under the control of its promoter rescued the learning defect of *capa-1* mutants on a NaCl gradient (Fig. 5e and Supplementary Fig. 7d). To further validate ASG as the cellular focus of peptide release modulating learned salt avoidance, we knocked down expression of the *unc-31* gene, essential for dense core vesicle release[46]. Animals expressing the *ets-5p::unc-31(RNAi)* transgene, silencing *unc-31* in ASG and

BAG[47], displayed a clear defect in NaCl-conditioned behavior compared to wild-type animals. Likewise, knockdown of *capa-1* in ASG using an *ops-1p::capa-1(RNAi)*[48,49] transgene caused a learning defect, while knockdown of both *capa-1* and *unc-31* did not result in an additive defect (Fig. 5f and Supplementary Fig. 7e). These results suggest that release of CAPA-1 peptides from ASG is required for gustatory aversive learning.

We previously reported that one of the CAPA-1 peptides, CAPA-1-3, activates NMUR-2[25]. Although *nmur-2* single mutants were not defective in gustatory aversive learning (Fig. 1d), NMUR-1 signaling may mask a role of this receptor in the modulation of salt chemotaxis behavior. The learning defect of a double mutant for the two receptors, however, resembled that of the *nmur-1* single mutant (Supplementary Fig. 8a–d), suggesting that only NMUR-1 mediates CAPA-1 signaling in gustatory aversive learning.

**CAPA-1 signaling is not involved in food searching behaviors**. NMU-like peptides are potent regulators of feeding[14,16], and *C. elegans* NMUR-1 modulates lifespan depending on the type of bacterial diet[28,50]. In addition, ASG neurons expressing CAPA-1 regulate foraging behaviors[47]. This prompted us to investigate if CAPA-1/NMUR-1 are involved in signaling the environmental food context and if this function could underlie its effect on gustatory aversive learning.

Loss of NMUR-1 extends lifespan only in worms grown on B-type *E. coli* strains such as OP50, but it has no effect in worms grown on K-12 type bacterial strains like HT115[28]. When assaying gustatory plasticity, we found that mutants of *capa-1* and *nmur-1* grown on HT115 display learning defects similar to those cultured on OP50 (Fig. 3b–e, Supplementary Figs. 7 and 8e–h). Thus, the effect of CAPA-1 signaling on gustatory aversive learning does not depend on bacterial diet.

We also tested whether CAPA-1/NMUR-1 signaling is involved in foraging behaviors. Upon removal from their *E. coli* food source, *C. elegans* display local search behavior with frequent reorientations, which are reduced in ASG-ablated animals[51]. Mutants of *capa-1* and *nmur-1* behaved like wild-type animals after removal from OP50 or HT115 (Supplementary Fig. 9a–f). Conversely, ASG neurons also modulate foraging when worms are feeding on *E. coli* bacterial lawns. On food, *C. elegans* spontaneously alternate between sedentary behavior, called dwelling, and active exploratory behavior, called roaming[52,53] that is promoted by ASG[47]. To determine if *capa-1* and *nmur-1* regulate foraging on food, we quantified the relative time that mutants spent roaming or dwelling and calculated mean speeds. Our results show that foraging is not affected in *capa-1* or *nmur-1* mutants feeding on OP50 or HT115 bacteria (Supplementary Fig. 9g–j). Taken together, these findings suggest that CAPA-1 signaling is not involved in general food sensing, because *capa-1* and *nmur-1* mutants display normal food searching behaviors.

**NaCl conditioning sculpts ASG calcium responses to salt**. Since CAPA-1 neuropeptides do not seem to be required for sensing salt or food conditions, we hypothesized that CAPA-1 signaling from ASG modulates neural processing of salt cues after aversive conditioning. Consistent with this model, laser ablation and calcium imaging studies suggest only a minor role for ASG in salt sensing[26,54,55], while ASG has been implicated in modulating behavioral responses to NaCl under starvation[56] and hypoxic stress[57].

To determine if NaCl conditioning influences ASG activity, we compared salt-evoked calcium responses in mock- and NaCl-conditioned wild-type animals using the calcium indicator GCaMP3[54,58]. Previous work showed that ASG responds to a 0–50 mM increase in salt concentration[54]. NaCl-conditioned

animals experience similar concentrations in our gradient assay, as worms are released around 50 mM NaCl at the center of the gradient and then migrate away from this concentration. Therefore, we imaged ASG calcium activity in response to a NaCl up- or downshift between 0 and 50 mM NaCl. Before imaging, animals were conditioned following the protocols for behavioral assays, i.e. we pre-exposed worms to a buffer with (NaCl-conditioned) or without (mock-conditioned) 100 mM NaCl for 15 min. ASG neurons showed small calcium responses to NaCl up- and downshifts in both mock- and NaCl-conditioned animals (Fig. 6a). The response to a downshift in NaCl concentration was significantly stronger after aversive conditioning compared to responses in mock-conditioned animals (Fig. 6a). CAPA-1 neurons thus respond differently to salt stimuli after aversive conditioning.

To probe effects of *capa-1* signaling further downstream of ASG, we asked if CAPA-1 neuropeptides modify NaCl-evoked calcium responses in AFD neurons where NMUR-1 is required for gustatory aversive learning. Using the calcium indicator GCaMP6s[59], we imaged calcium responses of AFD to NaCl up- and downshifts in mock- and NaCl-conditioned animals (Fig. 6b). Conform to previous work, AFD neurons responded to the onset and removal of salt stimuli[60,61]. Although NaCl conditioning evoked subtle changes in AFD responses to salt, the responses in NaCl-conditioned animals were not significantly different from those in mock-conditioned worms (Fig. 6b). NaCl-evoked AFD activity was also similar between wild-type animals and *nmur-1* mutants (Fig. 6b). Thus, NMUR-1 signaling does not seem to overtly affect NaCl-evoked calcium responses in AFD.

Next, we asked if NMU signaling adapts salt-evoked activity of the primary salt sensor, ASE. The ASEL(eft) and ASER(ight) pair exhibits functional asymmetry, with each partner activated by an increase and decrease in NaCl concentrations, respectively[55,62]. Consistent with this, mock-conditioned wild-type worms expressing the ratiometric YC2.12 indicator in ASE showed robust ASEL responses upon NaCl upshifts, while NaCl downshifts evoked an increase in calcium in ASER (Fig. 6c). NaCl conditioning reduced the responses of ASEL to NaCl upshift and sensitized ASER responses to NaCl downshifts (Fig. 6c), which is in agreement with previous reports[30,31]. ASE calcium responses to salt in mock- and NaCl-conditioned animals were unaffected in *nmur-1* mutants, indicating that NMU signaling is not required for the plasticity of ASE responses during gustatory aversive learning.

**CAPA-1 neurons mediate retrieval of learned salt aversion**. To gain further insight into how NMU-like neuropeptides regulate aversive learning, we investigated the temporal requirement of CAPA-1 neurons in the learning circuit. First, we chemically silenced ASG by expressing tetanus toxin light chain (TeTx), which impedes synaptic and dense core vesicle release[63], specifically in these neurons. As expected, ASG-specific TeTx expression impaired gustatory aversive learning, whereas it did not affect NaCl chemotaxis of mock-conditioned animals (Fig. 7a–d).

To further determine when ASG signaling is required, we optogenetically silenced these neurons during NaCl-conditioning or during the immediate recall of learned avoidance behavior when worms navigate a NaCl gradient. For this, we expressed the outward-directed proton pump Arch under control of the *capa-1* promoter[64] (Fig. 7e). Upon adding the essential cofactor for opsin activity (*all-trans* retinal, ATR), Arch can be activated by illuminating transgenic worms with green-red light (see Methods), which silences the host neuron. We varied the timing of illumination so that CAPA-1 neurons were silenced either during

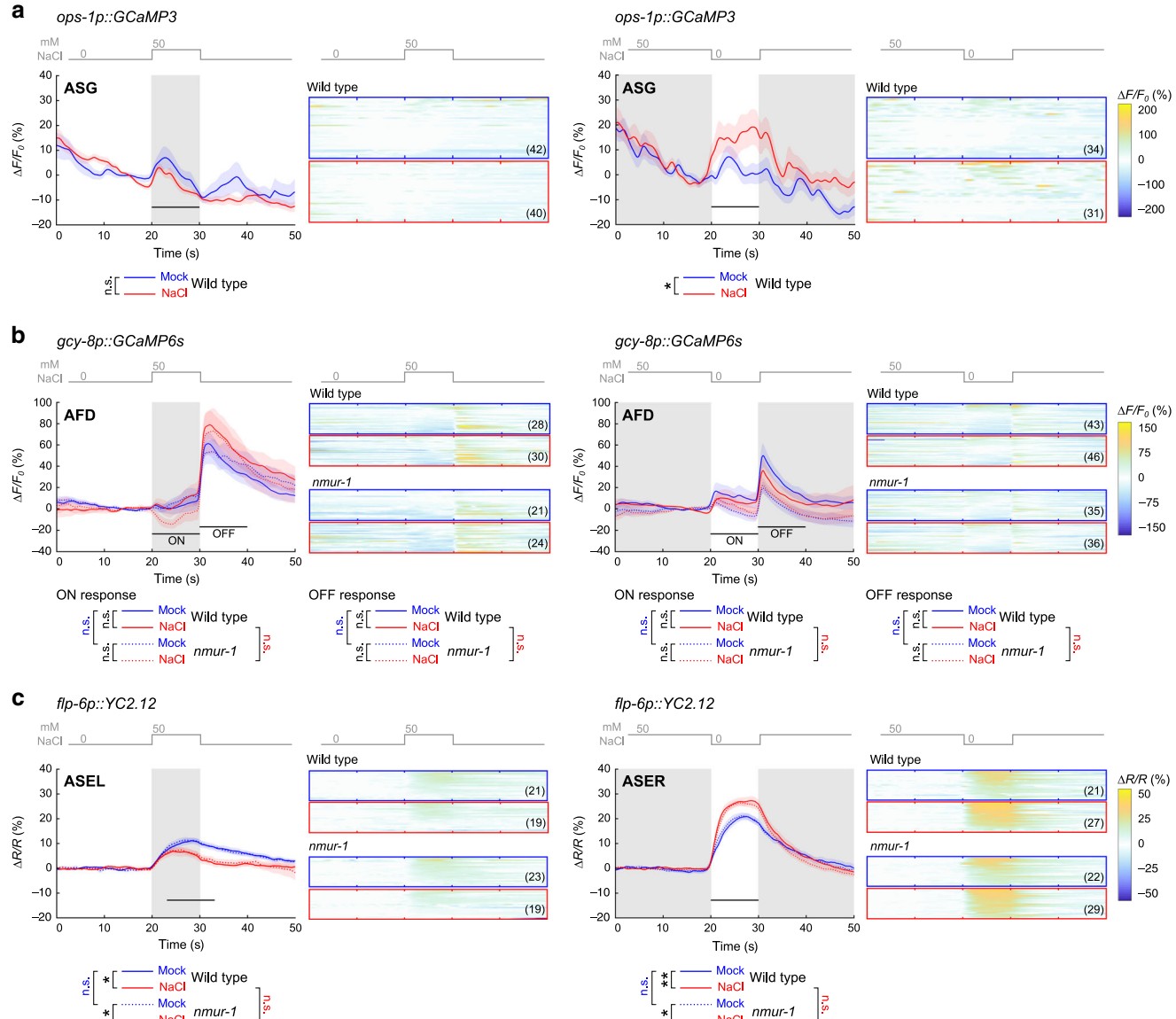

**Fig. 6 Aversive NaCl conditioning sculpts calcium responses of ASG neurons to salt stimuli.** NaCl-induced calcium responses of selected neurons in mock- and NaCl-conditioned animals. Animals are challenged to a 0–50 mM NaCl upshift (left panels) or a 50 to 0 mM NaCl downshift (right panels) after pre-exposure to 0 mM NaCl (mock-conditioned, blue traces) or 100 mM NaCl (NaCl-conditioned, red traces) for 15 min without food. Gray shading indicates exposure to 50 mM NaCl while shaded areas around the average traces indicate the S.E.M. Black horizontal bars show the time interval used for statistical comparison. Heat maps display all individual calcium traces for each condition. Number of animals pictured on each heat map. **a** NaCl-induced calcium responses in ASG neurons of mock- and NaCl-conditioned wild-type animals measured using the GCaMP3 indicator[54]. Calcium dynamics are relative to 10 s pre-stimulus control values, $\Delta F/F_0$. Significances determined by two-tailed Mann–Whitney $U$ test. n.s. not significant; *$p \leq 0.05$. **b** NaCl-induced calcium responses in AFD neurons of mock- and NaCl-conditioned wild-type or *nmur-1* (lst1672) animals expressing the GCaMP6s calcium indicator in AFD[59]. Calcium dynamics are relative to 10 s pre-stimulus control values, $\Delta F/F_0$. Significances determined by two-sided Kruskal–Wallis test. n. s. not significant. **c** NaCl-induced calcium responses in ASE neurons of mock- and NaCl-conditioned wild-type or *nmur-1* (ok1387) animals expressing the YC2.12 indicator in ASE[62]. Traces indicate the average percent change in the ratio of the distinct wavelengths of the YC2.12 indicator, $\Delta R/R$. Significances determined by one-way ANOVA with Tukey's multiple comparison test. n.s. not significant; *$p \leq 0.05$; **$p \leq 0.01$.

the acquisition of salt aversion or during retrieval of the learned behavior when navigating a NaCl gradient (Fig. 7f). To keep animals centered in the light beam for optogenetic silencing, we used smaller chemotaxis plates with a 0 to 50 mM NaCl gradient and tracked salt chemotaxis behavior during a 10-min time window (Fig. 7f). Silencing CAPA-1 neurons during the acquisition phase had little effect on learned salt avoidance (Fig. 7g, h and Supplementary Fig. 10a, b). In contrast, silencing during the retrieval phase significantly attenuated salt avoidance (Fig. 7g, h and Supplementary Fig. 10a, b). Optogenetically inhibiting CAPA-1

neurons had no effect on general locomotion in the absence of a NaCl gradient (Supplementary Fig. 10c, d). These results suggest that CAPA-1 neurons are required for expressing learned salt avoidance, but not for acquiring the conditioned response.

## Discussion

Neuropeptides are increasingly recognized as modulators of learning and memory, but how and when specific neuromodulators operate in learning circuits is not well understood. Here, we

 **9**

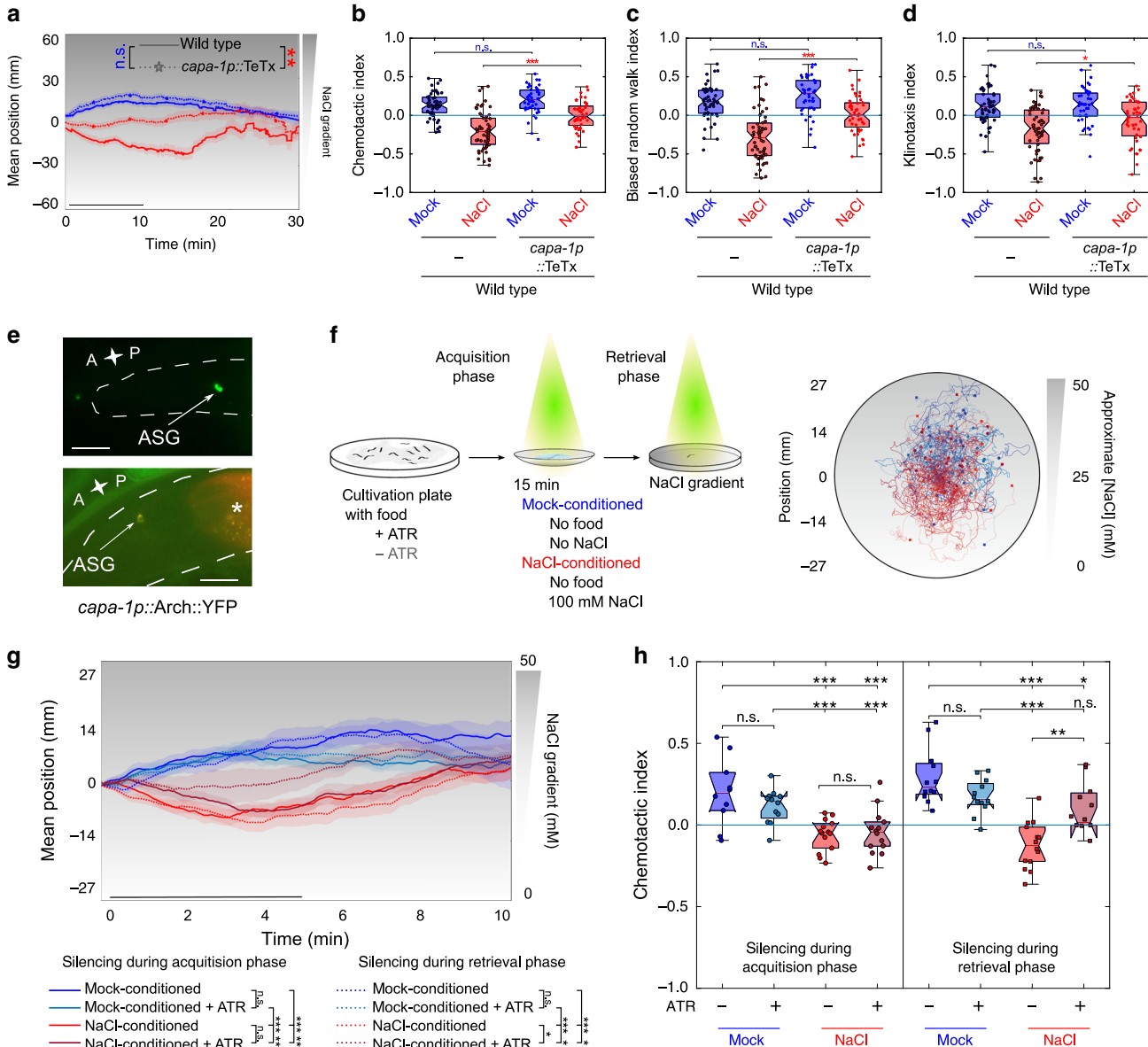

**Fig. 7 Silencing CAPA-1 neurons during retrieval of learned salt aversion attenuates NaCl-conditioned behavior. a–d** Mock- and NaCl-conditioned behavior of animals expressing the tetanus toxin light chain (TeTx) in ASG, which blocks synaptic release[63] (*capa-1p::TeTx::mCherry*). **a** Mean population position, **b** chemotactic, **c** biased random walk, and **d** klinotaxis indices on a 0–100 mM NaCl gradient after mock- or NaCl-conditioning. From left to right $n = 51, 59, 43, 44$ animals per condition. Results confirmed with a second independent strain (not shown). **e** The inhibitory opsin Arch expressed as Arch:: YFP under control of the *capa-1* promoter (top panel) co-localizes with ASG-specific mCherry expression[43] (bottom panel). Scale bar, 20 μm. **f** Optogenetic silencing of ASG. Individual worms are illuminated with yellow–green light for silencing CAPA-1 neurons during conditioning (acquisition phase) or during the NaCl chemotaxis assay when worms are navigating a linear NaCl gradient (retrieval phase) (see Methods for details). Arch requires supplementation of the cofactor all-trans retinal (ATR) to the worm culture. Transgenic *C. elegans* not fed ATR-supplemented food serve as a negative control. After conditioning, worms are put in the middle of a chemotaxis plate with a linear NaCl gradient (0–50 mM) and kept in the field-of-view using an automated xy-stage. Right panel shows navigation trajectories of 10–14 worms per condition. **g** Positions of individual worms on the NaCl gradient through time and **h** chemotactic index. Corresponding biased random walk and klinotaxis indices shown in Supplementary Fig. 10a, b. The time interval for statistical comparison was reduced to 5 min to accommodate the small plate format. **g, h** From left to right $n = 10, 12, 14, 14, 12, 12, 14, 11$ animals per condition. Shaded regions on **a** and **g** represent S.E.M. while black bars indicate the time interval for calculating the mean gradient position and behavioral indices. Significances determined by two-way ANOVA with Tukey's HSD post hoc test. n.s. not significant; *$p \leq 0.05$; **$p \leq 0.01$; ***$p \leq 0.001$. Boxplots show medians, 25th and 75th percentiles as box limits. The whiskers extend to the most extreme data points not considered as outlier.

discover a neuromodulatory role for NMU-like neuropeptides in an associative learning circuit in *C. elegans*. We show that CAPA-1 NMU-like neuropeptides, released from ASG sensory neurons, signal via their receptor NMUR-1 to specifically mediate the retrieval of learned salt avoidance.

The behavioral phenotypes that we observe highlight a neuromodulatory role of NMU-like signaling in gustatory aversive

learning. Several lines of evidence support this function. Mutants of *capa-1* and *nmur-1* are defective in different paradigms for gustatory aversive learning that rely on different types of aversive experience and salt gradients. We show that loss of *capa-1* and *nmur-1* mediate these phenotypes by disrupting aversive learning rather than causing general defects in salt sensing, osmotic homeostasis or detection of food-related cues. Underlying its role

in learned salt aversion, we find that CAPA-1/NMUR-1 signaling is required for aversive conditioning to adapt two elementary behavioral strategies, biased random walk and klinotaxis, in salt chemotaxis behavior. Thus, NMU-like neuropeptide signaling coordinately modulates distinct locomotory patterns that are required for learned salt avoidance, rather than specifically affecting a neural substructure of the conditioned response. Our findings are in agreement with recently reported functions of ASG in the modulation of locomotory mechanisms for experience-dependent salt chemotaxis[56].

By optogenetically manipulating ASG signaling, we show that CAPA-1 neurons mediate the retrieval of learned avoidance behavior, but do not appear to be required for the acquisition of this response. Evidence for such a temporally defined role of neuropeptide signaling in learning circuits also emanates from *Drosophila* studies on neuropeptide F and insulin-like peptides[65,66]. In *C. elegans*, knockout of the insulin receptor, using a conditional *daf-2* allele, has been shown to differentially affect the acquisition and retrieval of olfactory memory[67]. A defined temporal role of neuropeptides may be a general feature of their actions in learning circuits.

Aversive conditioning alters NaCl-evoked calcium responses in the *C. elegans* gustatory circuit[30,31]. For ASG neurons, we find that conditioning sensitizes calcium responses to a downshift in NaCl concentrations. Although the behavioral effects of this plasticity remain unclear, ASG activity has been correlated to turning behavior[56], and increased responses to NaCl downshifts could contribute to directed movements towards low salt within learned salt avoidance. We show that release of CAPA-1 neuropeptides from ASG modulates salt chemotaxis behavior upon aversive conditioning. As mock-conditioned behavior is not affected by NMU-signaling dysfunction, these results suggest a model in which CAPA-1 signaling has no primary role in innate salt attraction but is recruited upon aversive conditioning to modulate the gustatory circuit (Fig. 8). This finding is reminiscent of previous reports demonstrating that ASG modulates salt chemotaxis behavior under different stressful conditions, following hypoxia[57] and starvation[56].

Synthetic CAPA-1 neuropeptides activate the NMU receptor ortholog NMUR-1. In addition, we show that NMUR-1 is a cognate receptor for CAPA-1 peptides in vivo. NMUR-1 is expressed in several neurons including AIA[28], which is a cellular hub in the integration of aversive experience[27]. However, *nmur-1* does not seem to be required in these cells for learned salt aversion. We identify the AFD neurons as the main site where NMUR-1 facilitates gustatory aversive learning. While AFD has been well characterized for its role in thermosensation[68], it also responds to other sensory cues such as NaCl[60,61]. Aversive conditioning and loss of *nmur-1* does not significantly alter AFD calcium responses to salt. The mechanism by which NMU signaling modulates AFD activity upon NaCl conditioning therefore remains to be understood. NMUR-1 signaling may influence AFD's intrinsic excitability properties locally[31,69] or could alter AFD signaling in parallel or downstream of calcium transients[59,70]. AFD wires to an interconnected layer of sensory neurons and interneurons involved in biased random walk and klinotaxis strategies for salt chemotaxis behavior[23,24,29,33,39,71]. These include the primary ASE salt sensors that show experience-dependent calcium activity in response to NaCl[30,31]. Loss of *nmur-1* does not impair this experience-dependent plasticity, suggesting that AFD signaling may impinge on the gustatory circuit downstream of ASE to modulate salt chemotaxis behavior and change the NaCl valence from attractive to aversive. While we identify AFD as a major cellular focus for NMUR-1 in gustatory aversive learning, the possibility remains that other neurons expressing *nmur-1* also contribute to this behavior.

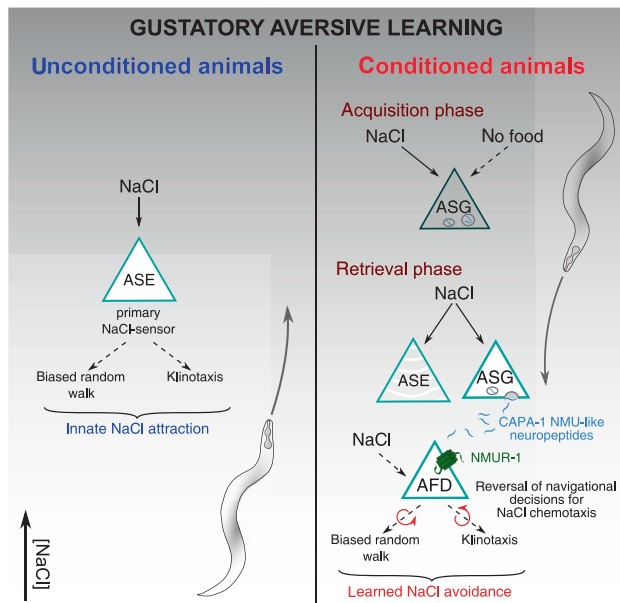

**Fig. 8 Schematic model of NMU signaling in gustatory aversive learning.** *C. elegans* displays innate NaCl attraction that is primarily mediated by the ASE salt sensors. This behavior involves both biased random walk and klinotaxis behavioral strategies that drive chemotaxis towards salt, for which NMU signaling is not required. Gustatory aversive learning induces aversion to salt. During training, the integration of NaCl and the aversive context of food absence engages ASG sensory neurons and alters ASG calcium responses to NaCl. Upon subsequent salt exposure during the retrieval of learned aversion, ASG releases CAPA-1 NMU neuropeptides which signal through their receptor NMUR-1 in AFD sensory neurons. NMU signaling mediates learned salt aversion by coordinately modifying biased random walk and klinotaxis behavioral choices during salt chemotaxis in accordance to previous experience.

The NMU signaling system has an ancient role in energy homeostasis across vertebrates and insects[14–18]. Despite its wide distribution in brain regions responsible for learning and processing of aversive experiences[14,15], only a few studies have postulated a role for mammalian NMU in memory based on its effects upon administration into the brain[20–22]. In *Drosophila*, the *capa* receptor gene is expressed in the mushroom bodies and neurons expressing the NMU precursor gene *hugin* are part of a sensory pathway that relays between gustatory neurons and higher brain centers[17,19,72,73]. *Hugin* may play a role in learning similar to *capa-1* in *C. elegans*, signaling aversive feeding conditions and stimulating motor programs facilitating the migration to favorable food environments.

Taken together, we have uncovered a neural mechanism by which the evolutionarily conserved NMU neuropeptide pathway mediates the retrieval of learned avoidance behavior. NMU-like peptides signal between sensory neurons that participate with primary salt sensors in the experience-dependent modulation of salt chemotaxis behavior. Neuropeptide-driven modulation of the gustatory circuit likely increases the sensory acuity of animals when faced with an aversive environment. Given that the NMU system is conserved across bilaterian animals, our study encourages further research into the functions of this neuropeptide system in other learning circuits.

## Methods

**C. elegans strains**. *C. elegans* was cultured using standard methods at 20 °C on Nematode Growth Medium (NGM) seeded with OP50 *E. coli* bacteria as a food

source, unless otherwise stated. Wild-type worms are of the Bristol variety N2. The following strains were acquired from the *Caenorhabditis* Genetics Center at the University of Minnesota: N2 wild type, RB1288 *nmur-1 (ok1387)*, RB2526 *nmur-2 (ok3502)*, VC1974 *nmur-3 (ok2295)*, RB2263 *capa-1 (ok3065)*, CX10 *osm-9 (ky10)*, GN122 *pgIs2 [gcy-8p::TU#813 + gcy-8p::TU#814 + unc-122p::GFP + gcy-8p:: mCherry + gcy-8p::GFP + ttx-3p::GFP]* and JPS271 *vxEx265 [gcy-8p::ICE + myo-2p::mCherry]*. *capa-1 (tm3764)* mutants were provided by the National Bioresource Project of Japan. Mutant genotypes were confirmed using PCR and backcrossed to N2 at least 6 times prior to analysis. All mutant and transgenic strains used in this study are listed in Supplementary Data 1.

**Phylogenetic tree**. To determine the relationship between NMU-like receptors from *Caenorhabditis elegans, Lottia gigantea, Drosophila melanogaster, Danio rerio* and *Homo sapiens*, a phylogenetic tree was generated by the maximum likelihood method using the phylogeny pipeline set in advanced mode with the number of bootstraps set to 100 (www.phylogeny.fr)[74]. The following sequences were used to construct the tree: *Homo sapiens* NMUR1 (AAH51914.1) and NMUR2 (EAW61653.1); *Danio rerio* Nmur-1a (ENSDARG00000060884), Nmur-1b (ENSDARG00000003944) and Nmur-2 (ENSDARG00000022570)[18]; *Drosophila melanogaster* capa receptor (Q8ITC7), pyrokinin-1 receptor (Q8ITC9), pyrokinin-2 receptor 1 and 2 (CG8784 and Q6NP98); *Lottia gigantea* NMUR (ULotgi1_91538)[7]; *Caenorhabditis elegans* NMUR-1 (Q18701), NMUR-2 (O17239) and NMUR-3 (Q7JNX1). Human (NP_003292.1), *D. rerio* (XP_687246.4, NP_001108160.1, and Q1MT85), *L. gigantea* (Lotgi1_122499)[7] and *C. elegans* (Q95YD7) sequences of the thyrotropin-releasing hormone receptor family were chosen as an outgroup to root the tree.

**Molecular biology**. All oligonucleotide primers used in this study are listed in Supplementary Data 1.

For heterologous expression of NMUR-1 in Chinese hamster ovary (CHO) cells, *nmur-1* cDNA was amplified by PCR from cDNA of mixed-stage wild-type *C. elegans* and directionally cloned into the eukaryotic expression vector pcDNA3.1 (Invitrogen).

GFP reporter and rescue constructs were generated from a modified pSM vector carrying a GFP reporter sequence preceded by an SL2 trans-splicing sequence (kindly provided by C. Bargmann, Rockefeller University, New York, USA). For *nmur-1*, the *nmur-1* cDNA was first amplified from mixed-stage wild-type *C. elegans* template and inserted into a pSM backbone (linearized by BamHI and NheI restriction digest) using Gibson assembly. The 1 kb *nmur-1* 3′UTR sequence was then inserted after the GFP coding sequence by Gibson assembly into the backbone digested by EcoRI and SpeI, which replaced the *unc-54* 3′UTR sequence of the original pSM plasmid. Finally, a 2 kb *nmur-1* promoter fragment was inserted prior to the gene's cDNA sequence after SphI and XbaI digestion.

For cell-specific rescue experiments, the *nmur-1::SL2::gfp::nmur-1 3′UTR* pSM backbone was first linearized by PCR amplification, yielding a 5900 bp linear sequence in which cell-specific promoters were inserted using Gibson Assembly. For this, promoter regions were amplified from genomic N2 DNA, with a sequence 2877 bp upstream of the predicted coding region for *gcy-28d*; 2648 bp for *odr-2*; 2393 bp for *osm-6*; 3000 bp for *ceh-23*; 2534 bp for *ocr-2*; 2009 bp for *gcy-8*; 2067 bp for *srh-142*; 2449 bp for *srh-220* and 4786 bp for *ocr-4*. In addition, a 314 bp *ins-1(s)p* was amplified from pCEC19 (kindly provided by C. Bargmann, Rockefeller University, New York, USA) and a 485 bp *flp-1(tr)p* from pCS178 (kindly provided by A. Gottschalk, Johann Wolfgang Goethe-Universität Frankfurt, Germany).

For *capa-1* expression, 2.5 kb of the *capa-1* 3′UTR sequence was inserted into the pSM backbone linearized by EcoRI and SpeI using Gibson assembly. Next, a genomic region encompassing 2.2 kb of the sequence upstream of the *capa-1* start codon and the *capa-1* genomic sequence were amplified from genomic DNA of mixed-stage wild-type *C. elegans* and inserted into the pSM backbone (digested with BamHI and NheI) by Gibson assembly.

For ASG silencing using Arch or tetanus toxin (TeTx), we inserted the 2.2 kb *capa-1* promoter sequence upstream of the Arch and TeTx coding sequences in pSH122[64] (linearized by SphI and MscI) and pSH188[64] (linearized by NotI and NheI), respectively.

Transgenes for ASG-specific *capa-1* knockdown were constructed following a reverse genetics approach[49]. Briefly, the *ops-1* promoter (1.9 kb) was fused to a 1132 bp genomic *capa-1* fragment in sense and antisense orientation. The genomic *capa-1* fragment is identical to that of the clone taken up in the Ahringer feeding RNAi library (https://www.sourcebioscience.com/life-science-research/clones/rnai-resources/c-elegans-rnai-collection-ahringer/).

The transgene for ASG-specific *gcy15p::mCherry expression* was constructed through fusion PCR. First, the 995 bp mCherry sequence was amplified from pGH8, and the 1908 bp promoter and 883 bp 3′UTR of the *gcy-15* gene was amplified from genomic DNA. All three fragments were sequentially fused through fusion PCR.

**CRISPR/Cas9-mediated *nmur-1* knockout**. The *lst1672* deletion allele of *nmur-1* was generated using a common CRISPR/Cas9 genome editing protocol[75]. Specific guide RNAs were designed using the Custom Alt-R® CRISPR-Cas9 guide RNA design tool from IDT (https://eu.idtdna.com/site/order/designtool/index/CRISPR_CUSTOM), thereby selecting 2 guide RNAs to generate a deletion closely

resembling the *nmur-1 (ok1387)* deletion. A co-CRISPR strategy was used based on *dpy-10* as a marker to enrich for genome-editing events[75]. The mix consisted of 2.5 μL recombinant *C. elegans* codon-optimized Cas-9 protein (15 mg/mL, kindly provided by the Hollopeter lab, Cornell University), 2.5 μL tracrRNA (0.17 mol/L, IDT), 1 μL *dpy-10* crRNA (0.6 nmol/μL, IDT), 0.5 μL *nmur-1* crRNA1 (0.6 nmol/μL, IDT), 0.5 μL *nmur-1* crRNA2 (0.6 nmol/μL, IDT), 1 μL *dpy-10* repair template (0.5 mg/mL, Merck) and 1 μL repair template *nmur-1* (1 mg/mL, IDT) and was injected into the gonads of young adults. F1 transgenic worms were transferred to individual NGM plates and allowed to produce F2 progeny after which F1 progeny was screened by PCR using specific primers. Sequences of the crRNAs and repair templates can be found in Supplementary Data 1.

**In vitro GPCR activation assay**. The GPCR activation assay was performed following our established protocol[34]. Mammalian CHO K1 cells stably overexpressing apo-aequorin and human $G\alpha_{16}$ were transiently transfected with the *nmur-1/* pcDNA3.1 or empty pcDNA3.1 (control) plasmid using Lipofectamine LTX and Plus reagent (Invitrogen). After transfection, cells were grown overnight at 37 °C, after which they were transferred to 28 °C and allowed to incubate for 24 h. On the day of the assay, CHO cells were harvested from their culture flask and loaded with Coelenterazine H (Invitrogen) for 4 h at room temperature, which elicits the formation of the calcium-sensitive photoprotein aequorin. The incubated cells were then added to synthetic peptides dissolved in DMEM/BSA, and luminescence measured for 30 s at 496 nm using a Mithras LB940 luminometer (Berthold Technologies). After 30 s of readout, 0.1% triton X-100 was added to lyse the cells, resulting in a maximal calcium response that was measured for 10 s. After initial screening, concentration-response curves were constructed for HPLC-purified CAPA-1 peptides by subjecting the transfected cells to each peptide in a concentration range from 0.1 nM to 100 μM. Cells transfected with an empty vector were used as a negative control. Assays were performed in triplicate on at least two independent days. Concentration-response curves were fitted using GraphPad Prism 5 (nonlinear regression analysis with a sigmoidal concentration-response equation).

A peptide library of over 300 synthetic *C. elegans* peptides that was used to challenge the NMUR-1 expressing CHO cells was compiled based on in silico predictions and peptidomics data. Peptides were synthesized by Thermo Scientific and GL Biochem Ltd.

**Transgenesis and expression pattern analysis**. To generate transgenic *C. elegans*, constructs were injected into the syncytial gonad of young adult worms at concentrations ranging from 5 to 50 ng/μL with 50 ng/μL of the coinjection marker *elt-2p::GFP* or *unc-122p::dsRED* and 17 ng/μL of a 1-kb DNA ladder (Thermo Scientific) as carrier DNA.

Expression patterns of *capa-1* transgenes were visualized by an inverted Zeiss AxioObserver Z1 microscope fitted with a 40X oil objective, an ORCA-Flash4.0 V2 camera (Hamamatsu) and W-View GEMINI image splitting optics (Hamamatsu). Image acquisition was performed using Metamorph (Molecular Devices) software. Expression of *capa-1* in ASG neurons was confirmed by co-localization with mCherry expressed under control of the ASG-specific *gcy-15* promoter[43] for two independent transgenic strains: a reporter driving expression of the *capa-1* gDNA under control of its endogenous promoter and 3′UTR, and an Arch::YFP translational fusion under control of the *capa-1* promoter. Fluorescence was visualized in at least five animals for each independent strain. The expression of *nmur-1* was visualized on an Olympus FluoView FV1000 (IX81) confocal microscope. Expression was visualized in at least 5 animals for two independent strains. For imaging, hermaphrodite animals were immobilized on 10% agarose pads using polystyrene beads between pad and coverslip to restrain the animal's movement (Polybead® Microspheres 0.10 μm, Polyscience). Expression patterns were confirmed in at least two independent transgenic strains.

**Quadrant-based salt chemotaxis assays**. Salt chemotaxis and gustatory aversive learning were assessed on quadrant plates following an established protocol[34,36]. All behavioral assays were performed in a climate-controlled room set at 20 °C and 40% relative humidity. In brief, 1-day young adult hermaphrodites were grown at 25 °C on culture plates seeded with sufficient *E. coli* OP50 or HT115.

For salt chemotaxis toward increasing NaCl concentrations, worms were collected from their culture plate and washed three times over a period of 15 min with chemotaxis buffer (CTX, 5 mM $KH_2PO_4/K_2HPO_4$ pH 6.6, 1 mM $MgSO_4$, and 1 mM $CaCl_2$). Salt chemotaxis behavior was then tested by pipetting 100 – 200 animals on the intersection of four-quadrant plates (Falcon X plate, Becton Dickinson Labware) filled with buffered agar (2% agar, 5 mM $KH_2PO_4/K_2HPO_4$ pH 6.6, 1 mM $MgSO_4$, and 1 mM $CaCl_2$) of which two opposing pairs have been supplemented with NaCl (100, 200, 300 or 400 mM NaCl). Assay plates were always prepared fresh and left open to solidify and dry for 60 min. Plates were then closed and used on the same day. After animals were allowed to crawl for 10 min on the quadrant plate, a chemotaxis index was calculated as $(n(A) - n(C))/(n(A) + n(C))$ where $n(A)$ is the number of worms within the quadrants containing NaCl and $n(C)$ is the number of worms within the control quadrants without NaCl.

For gustatory aversive learning, well-fed synchronized adult worms were washed in CTX buffer with or without 100 mM NaCl for NaCl-conditioned and

mock-conditioned worms, respectively. Salt chemotaxis behavior was then assayed on four-quadrant plates that contain 25 mM NaCl in one pair of opposing quadrants. After 10 min, the distribution of worms over the quadrants was determined and a chemotaxis index calculated as described above.

In chemotaxis assays using glycerol conditioning, worms were washed for 15 min with CTX buffer in which NaCl has been replaced with glycerol at the same osmolarity (200 mM). To assess if NaCl conditioning affects osmotic preference, chemotaxis to glycerol was tested on four-quadrant plates that contain 50 mM glycerol in the two opposing quadrants of the assay plate.

In salt chemotaxis assays pairing NaCl with food[32], adult animals were washed off culture plates using CTX buffer with or without 100 mM NaCl and allowed to sediment for a few min. They were then transferred to plates filled with buffered agar (2% agar, 5 mM $KH_2PO_4/K_2HPO_4$ pH 6.6, 1 mM $MgSO_4$, and 1 mM $CaCl_2$) with or without 100 mM NaCl that were unseeded or seeded with 200 μL of 0.5 g/mL freshly grown OP50 bacteria in $H_2O$. After 30 min of conditioning on plates, animals were washed off again using CTX buffer ± NaCl, allowed to sediment for 1 min, and placed on a chemotaxis quadrant plate (25 mM NaCl) to measure salt chemotaxis behavior.

For salt chemotaxis assays with benzaldehyde conditioning, 6 μL of undiluted benzaldehyde was pipetted on the lid of seeded plates (with or without 100 mM NaCl) just prior to transferring worms, after which the plate was immediately sealed with Parafilm[32].

**Quantification of salt chemotaxis on linear NaCl gradients**. Tracking of salt chemotaxis behavior on salt gradients was performed on 0–100 mM linear NaCl gradients[39], unless otherwise stated. NaCl gradients were generated in square Petri dishes (Greiner, 120 × 120 × 17 mm, vented) by elevating one side of the plate and filling half with 50 mL of buffered agar (2% agar, 5 mM $KH_2PO_4/K_2HPO_4$ pH 6.6, 1 mM $MgSO_4$, and 1 mM $CaCl_2$) supplemented with 100 mM NaCl. The agar was allowed to solidify for 30 min, creating a triangle wedge. The plate was then laid flat to fill the other half with 50 mL buffered agar without NaCl. Behavioral assays were performed 24 hr after the plates were generated. Before the start of the assay, 15–20 young adult worms were conditioned for a period of 15 min using CTX buffer with 100 mM NaCl (NaCl-conditioned) or NaCl-free CTX buffer (mock-conditioned). Animals were then pipetted at the center of the square assay plate corresponding to approximately 50 mM NaCl.

Assay plates were imaged using an in-house tracking platform consisting of 10 MP NET GP11004M cameras fitted with LM16JC10M KOWA lenses. A diffuse LED light source (Rosco 12"x12" LitePad) was used to illuminate animals in trans. Two consumer privacy filters (3 M PF17.0) were stacked perpendicularly on the glass stage holding the assay plates to enhance contrast. StreamPix 6 multicamera software (Norpix, Montreal, Canada) was used to acquire footage at two frames per second. Because behavioral assays are sensitive to ambient conditions, care was taken to record experimental and control conditions simultaneously by dedicated cameras.

The resulting footage was analyzed using custom particle-tracking algorithms written in MATLAB (Mathworks, Natick, MA), built upon the algorithm of a previously reported script available at https://sourceforge.net/projects/wormtracker/[76]. These scripts are available upon request. Briefly, worms were identified by average background subtraction, gaussian smoothing of the resulting foreground image and intensity thresholding to binarize each frame. Blobs falling within a predefined size range for single animals were connected over adjacent frames into individual tracks. Coordinates of the center of mass (centroid) and simple shape parameters were saved for each worm object on each frame. Individual worm tracks terminate when worms collide with each other or with the boundaries of the behavioral arena. The complete trajectory of an individual worm throughout the recording can therefore be represented by multiple tracks of varying length. Short tracks with a total duration of less than 5 min were discarded. The remaining trajectories were smoothed with a 1.5 s moving average filter before calculating instantaneous speed of the centroid.

The movement of each animal's center of mass was considered to automatically label different behavioral states. Pauses were labelled when the centroid's velocity falls under 0.01 mm/s for more than half of all time points in a 10 s sliding time window. The animal's behavioral state was classified as a turn if, for any given time point, the geometric angle between the points behind or ahead the track by 0.3 mm is <80°[29]. Any time points after the turn in which the velocity remains smaller than 0.1 mm/s were also classified as turn[77]. According to this definition, reversals and omega turns were both recognized as turns[51]. When individual turns are separated by <3.8 s, they were bundled into a pirouette[29].

To quantify NaCl attraction of mock- and NaCl-conditioned worms on linear NaCl gradients, several indices were computed. For each frame in the 30-min recording after release on the linear NaCl gradient, the position of each individual worm was determined and the population mean position traced through time. For statistical comparison of the mean individual position and for quantifying behavioral indices (see below), we chose to account for experiment-to-experiment variability by defining a time interval in which behavior was quantified (specified by a black horizontal bar in the mean position plot). This time interval usually spanned the first 10 min from the start of the experiment, which is the time interval in which the magnitude of chemotactic migration was the largest. Tracks active less than 5 min within this 10-min interval were also discarded, hereby preventing

multiple tracks from one animal to be included in the analysis. For tracks meeting these requirements, the average position was first computed for each worm. Next, the chemotactic index, which reflects chemotactic rate and direction, was quantified for each individual animal trajectory by dividing the mean velocity along the gradient direction $\langle v_g \rangle$ by the mean crawling speed along the trajectory $\langle s \rangle$[39]. This index is positive when an individual worm predominantly moves up the gradient, and negative when it migrates to lower salt concentrations. To quantify the biased random walk index, trajectory stretches in which worms exhibiting run behavior were extracted, and the fractional difference of run duration up or down the gradient was calculated for each worm ($(\langle \text{length run}_{up} \rangle - \langle \text{length run}_{down} \rangle)/(\langle \text{length run}_{up} \rangle + \langle \text{length run}_{down} \rangle)$)[39]. The klinotaxis index was determined by the fractional difference in the probability of run behavior after a reorientation event (turn or pirouette) to be initiated up or down the gradient ($(\langle P(\text{reorient}_{up}) \rangle - \langle P(\text{reorient}_{down}) \rangle)/(\langle P(\text{reorient}_{up}) \rangle + \langle P(\text{reorient}_{down}) \rangle)$)[39]. For this, a stretch of 5 s from the beginning of each run was considered. Runs were discarded when another behavior was initiated within 5 s.

**Measuring Cl⁻ in linear NaCl gradients**. The actual concentrations of chloride ion were measured using a chloride electrode (perfectION$^{TM}$ – Mettler-Toledo) to verify the linear relationship of NaCl concentrations along the y-axis in linear NaCl gradients. In all, 24 h after pouring agar plates only supplemented with NaCl, 1 cm wide samples along the y-axis were excised from the agar slab. Samples were solubilized by microwave heating in ultra-pure water to an end volume of 10 mL and were kept at 60 °C in a water bath. The electrode was calibrated using a serial dilution of Chloride Standard Solution (1000 mg/L Cl⁻, NIST®) at 60 °C. In all, 200 μL ionic strength adjuster (ISA solid state ISE, Mettler-Toledo) was added to the sample prior measurement according to the manufacturer's protocol.

**Locomotion on a bacterial lawn and after removal from food**. For quantitative analyses of local search behavior and roaming or dwelling, worms were grown until the first day of adulthood at a low population density, on NGM plates seeded with OP50 or HT115, in a climate-controlled room (20 °C and 40% relative humidity).

For assays on food, assay plates were prepared one day prior to the experiment by seeding freshly poured 90 mm diameter Petri plates with 500 μL of fresh OP50 or HT115 culture and allowing the bacteria to grow overnight in the climate-controlled room. On the day of the assay, 5–10 animals were first picked to a holding plate prepared in parallel with the assay plates. After 45 min, animals were moved to the assay plate and placed on the imaging platform and recorded for 90 min after 30 min of acclimatization. Worms were tracked from this footage as described above. Tracks shorter than 20 min were discarded. To classify behavior into roaming and dwelling, speed and angular speed (a measure of path curvature) were averaged over 10 s intervals. Both classes were separated by tracing a line (Speed = Curvature/450) through the plot scattering speed in function of curvature. Time points falling above the line were considered roaming intervals, while those below the line were labelled as dwelling[52,53]. For each animal, the fraction of time in each behavioral state was computed, as well as each individual's average speed during dwelling or roaming.

For local search behavior off food, 5–10 well-fed worms were picked from their culturing plate to an unseeded intermediate plate and within 1 min transferred to another unseeded plate on which they were filmed for 15 min[51]. Mean speed was calculated per individual, as well as the fraction of time spent in each behavior (pause, run, turn and pirouette) as determined by the segmentation algorithm explained above.

**Calcium imaging in microfluidic chip**. Transgenic worms expressing GCaMP3 in ASG[54], YC2.12 in ASE[62] or GCaMP6s in AFD[59] were imaged in a microfluidic polydimethylsiloxane (PDMS) chip[58]. The chip is attached to inlet tubes, connecting the worm-loading channel to a reservoir of CTX buffer (with or without 100 mM NaCl). Well-fed transgenic worms were first picked in the reservoir where they were exposed to CTX buffer in the absence of food for approximately 15 min. NaCl-responses were then assayed by positioning the worm in the channel and challenging the animal with a 0 to 50 mM NaCl upshift or a 50–0 mM NaCl downshift. The composition of buffers for conditioning and imaging is the same as buffers used in behavioral assays, except for the addition of sucrose to balance osmolarity (5 mM $KH_2PO_4/K_2HPO_4$ pH 6.6, 1 mM $MgSO_4$, and 1 mM $CaCl_2$ and 0, 50 or 100 mM NaCl and adjusted to 350 mOsmol/kg$H_2O$ by sucrose). Imaging was conducted on an inverted Zeiss AxioObserver Z1 microscope. Fluorescent images were captured using a 40X objective and an ORCA-Flash4.0 V2 camera (Hamamatsu) driven by Metamorph software (Molecular Devices). Images were analyzed using custom-written Mathematica (Wolfram) code setting a region of interest around the cell body (available upon request). For GCaMP imaging, the fluorescence intensity of the neuron was subtracted by the median background intensity signal of the entire channel stream and the fluorescence intensity during the 10 s time window before stimulus delivery was averaged and defined as $F_0$. $\Delta F/F0$ (%) is calculated as $100 \times (\text{background corrected fluorescence} - F_0)/F_0$. For YC2.12 imaging, $\Delta R/R$ (%) is calculated as the ratio between CFP and YFP fluorescent emissions.

**Optogenetic silencing of ASG**. Transgenic worms expressing the outward-directed proton pump Arch in ASG were cultured at 25 °C on culture plates seeded with sufficient OP50 *E. coli*. One day before the assay, transgenic worms were transferred to OP50 culture plates to which *all-trans* retinal (ATR, Sigma Aldrich) has been added (300 μM final concentration in fresh OP50 liquid culture). This cofactor is needed for Arch activity, and allows transgenic worms transferred to culture plates without ATR to serve as negative control. A 0 to 50 mM linear gradient was generated in a 55 mm Petri plate. First, 15 mL buffered CTX agar supplemented with 50 mM NaCl was poured in a 55 mm diameter Petri dish elevated to one side. An additional 15.5 mL buffered CTX agar without NaCl was poured on top of the first triangular wedge once it solidified. NaCl was allowed to diffuse at least 24 h prior the assay. Individual worms were washed for 15 min in a small volume of CTX buffer with or without 100 mM NaCl (NaCl-conditioning or mock-conditioning). After conditioning the worm was transferred to the center of the 0 to 50 mM NaCl gradient plate, mounted on a Zeiss AxioObserver Z1 microscope. The animal was then allowed to navigate the plate for 10 min while an automated xy-stage kept it in the field-of-view. For this, a low-resolution Stingray F-145B camera (Allied Vision) was mounted to the microscope, and repositioned the Prior ProScan III stage each second to center the worm in the field-of-view using MATLAB code adapted from Wang et al.[78]. Each animal's path was reconstructed based on the recorded low-resolution video-stream and the logged stage positions. The behavior was then segmented and indices reflecting chemotaxis behavior were calculated as described above. For ASG silencing during conditioning, the worm was kept under yellow–green light of a fluorescent stereomicroscope during the washing procedure (Leica MZ16F microscope and EL6000 external light source with mCherry Leica filter set, number 10450195 – 540 −580 nm; Intensity 0.2 to 0.4 mW/mm²). For silencing during the navigation period, the worm was continuously illuminated with 567–602 nm light while it was being tracked (Zeiss Colibri.2 LED module with Filter cube 61 HE GFP/HcRed, intensity ~0.552 mW/mm²).

**Quantification and statistical analysis**. Throughout this work, individual data points are scattered on boxplots of which the central mark indicates the median and the bottom and top edges the 25th and 75th percentile, respectively. The whiskers extend to the most extreme data points not considered as outlier (boxplots generated by MATLAB's boxplot() function). Outliers were always included in the statistical analysis and are represented on each figure. For plots showing the mean population position on linear NaCl gradients through time, shaded areas around the mean trace denote S.E.M. Mean shaded areas around the mean Ca²⁺-activity traces in Fig. 6 also represent S.E.M.

Detailed information on statistical analysis has been included in Figure legends. Briefly, data were first tested for normality using the Shapiro-Wilk normality test (alpha = 0.05). When comparing the effect of (neuro)genetic intervention to wild-type worms and appropriate controls under one condition (i.e. only comparing worms with identical treatment history such as only NaCl-conditioned animals), one-way ANOVA followed by Tukey's multiple comparison tests were performed. When one of the conditions failed to pass the Shapiro-Wilk normality test, a two-sided Kruskal–Wallis with Dunn's multiple comparison test was employed. For analyses in which different conditioning regimens are compared between genotypes, significances were computed by two-way ANOVA with Tukey's HSD (honestly significant difference) post hoc test. For ASG calcium imaging experiments, the response of mock- and NaCl-conditioned animals to NaCl pulses was compared using the nonparametric two-tailed Mann–Whitney U test (Fig. 6). For the fraction of time in each behavioral state, the relative fraction was binned over individual worms and analyzed for statistical significance by Chi-squared analysis (Supplementary Figs. 9c, f and 10d). Different levels of significance are indicated as n.s. not significant; *$p \leq 0.05$; **$p \leq 0.01$; ***$p \leq 0.001$.

For NaCl chemotaxis behavior on linear NaCl gradients, 20–30 worms are tracked per experiment. Wild-type, control and experimental conditions are simultaneously monitored with a multi-camera setup. From that footage, the mean position, chemotactic, biased random walk and klinotaxis indices are calculated for each individual trajectory (or track) of sufficient quality (i.e. longer than 5 min within the respective indicated time interval, detailed above). Data was collected on at least two independent days, which typically pooled 6–10 experimental repeats. Independent assays were compared to each other (with no significant changes unless noted), and the data pooled for analysis as a single group. This resulted in animal numbers allowing robust statistical analysis (sample size for tracking experiments are the number of analyzed worms, typically in the 50–100 animal range). Although unusual, all data from one experimental day was discarded when either wild type or control conditions on that day failed to conform to previous observations, which commonly could be traced back to abnormalities in ambient conditions. Similarly, all other behavioral assays were performed on at least two independent experimental days.

For experiments with transgenics harboring extrachromosomal arrays, animals with clear target fluorescence emanating from the pSM backbone, TeTx::mCherry or Arch::YFP fusions were selected as transgenic animals. If the strain was not injected with a specific fluorescent reporter, animals were selected based on their fluorescent co-injection marker. All experiments with transgenic animals holding an extrachromosomal array were performed with the un-injected background strain as a control.

Expression patterns were confirmed in at least five animals for two independent strains.

**Reporting summary**. Further information on research design is available in the Nature Research Reporting Summary linked to this article.

## Data availability
The data generated and/or analyzed during this study are available from the corresponding author on reasonable request. The source data underlying Figs. 1–7 and Supplementary Figs. 1–10 are provided as a Supplementary Data 2.

## Code availability
All the code used in this study is available upon request.

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

## Acknowledgements

We thank members of the Schoofs, Beets and Temmerman labs for their thoughtful comments. We would like to thank F. J. Naranjo-Galindo for technical assistance. We thank the *Caenorhabditis* Genetics Center, supported by the NIH Office of Research Infrastructure Programs Grant P40 OD010440, the Mitani lab and the National

Bioresource Project of Japan, the Alcedo lab (Wayne State University), the Bargmann lab (Rockefeller University), the Colón-Ramos lab (Yale University), the Hollopeter lab (Cornell University), the Jorgensen lab (University of Utah), the Murayama lab (OIST Graduate University), the Pocock lab (Monash University), S. Husson and the Gottschalk lab (Johann Wolfgang Goethe-Universität Frankfurt) and the Schafer lab (MRC Laboratory of Molecular Biology) for providing *C. elegans* strains and reagents. J.W., K. P., C.B., I.R. and I.B. are fellows of the Research Foundation - Flanders (FWO). This research was supported by European Research Council Grant 340318, KU Leuven Research Council Grant C14/15/049 and FWO Grant G093419N.

## Author contributions

J.W., K.P., P.V.d.A., and I.B. conceived and designed experiments. J.W., E.V., C.B., K.P., and S.V.D. performed molecular genetics and worm transgenesis. J.W. and K.P. performed all quadrant-based NaCl chemotaxis assays. J.W. performed all behavioral and optogenetic experiments. J.W. and P.V.d.A. conducted calcium imaging. I.R. and J.L. assisted with microfluidic chip construction. R. J. advised the construction of the imaging platform. J.W., P.V.d.A., K.P., and I.B. analyzed and interpreted results. J.W., K.P., L.S., and I.B. wrote the paper.

## Competing interests

The authors declare no conflict of interest.
