## [Peer Review File · Nature Communications]

Reviewers' Comments:

Reviewer #1:

Remarks to the Author:

In this paper, Watteyne et al contend that CAPA-1 / NMUR-1 peptidergic signaling is necessary for memory retrieval in *C. elegans* gustatory aversive learning. The authors begin by identifying a gustatory aversive learning defect in mutants lacking *nmur-1*, and further localize the role of *nmur-1* to a single neuron pair (AFD). After identifying CAPA-1 as the neuropeptide that signals to NMUR-1, they demonstrate that inhibiting chemical signaling from the CAPA-1 expressing neuron pair (ASG) impairs gustatory aversive learning. Further, the authors silence the ASG neuron pair during either acquisition or retrieval; this demonstrates that ASG neuronal activity is necessary for retrieval, but not acquisition, of gustatory associative learning. The evidence presented is convincing (with one exception, noted as a major revision below), and the identification of a conserved neuropeptide as a modulator of memory retrieval is a valuable contribution to the study of learning and memory. I expect that the findings of this paper will be of interest to a wide readership.

Major:

Overall, the data for the gene rescue experiments for *nmur-1* are poor. On the one hand, it is convincing that AFD is involved (given consistent results across three promoters). On the other hand, the variability of the WT control (which should be fairly stable) is at least as large, if not sometimes larger than, the genotype differences (e.g. compare WT chemotactic index between 3B and 3E, compared to the WT vs mutant chemotactic index in 3D). When compared to the initial WT and *nmur-1* mutant results (Fig. 2D-F), many of the "unsuccessful" rescues look more similar to WT than *nmur-1* mutants (e.g. supp. 2D, supp 2E). This creates some doubt regarding the gene rescue results.

Ideally, the authors should re-run these experiments using the original chemotaxis assay (Fig. 1D), which minimizes noise by using 100+ animals per assay. Additionally, using non-fluorescent mutant animals from co-injection experiments is a poor control. It allows for the possibility that some of the animals in the mutant group are "silent rescues" – worms with the rescue plasmid who did not get the co-injection fluorescence. I expect this may be the reason that, for instance, chemotactic index in Fig. 2I for *nmur-1* mutants is closer to the wildtype than the mutant in 2D, and that klinotaxis showed no difference between mutants and WTs in supp. 2A and supp. 3C. Ideally, the authors would repeat these experiments using an empty plasmid with the co-injection marker for mutant controls; if that isn't feasible, using un-injected mutant controls would at least show an equivalent mutant phenotype for comparison.

Several experiments (e.g. many of the mock groups in Fig. 1D, supp. 5A-B) have very small N – it looks like as low as 3. That is really too few. Off the cuff, I would suggest a minimum of 10. N should be added to any experiments with low N, and the authors should either be relatively conservative in making sure to include enough N, or include a power analysis to provide a basis for their decision to use low N. Additionally, it is not clear why there are sometimes drastic differences in the numbers of animals between experiments (e.g. Fig. 2D-F use "at least 25 animals", while the same assay in Fig. 2I uses "at least 284 animals"). This should be addressed.

Minor:

Fig. 1C has a typo – the denominator should be $n(A) + n(C)$, not -.

It would be nice to have a representative picture of the worms in the salt gradient assay (to allow the reader to assess the imaging setup – contrast, resolution, etc).

Several experiments use custom code created in the lab. It would be nice if that code would be packaged as part of the supplemental, to allow readers to check the code independently.

The authors should clarify how outliers were determined, and whether they were included in statistical analysis between groups.

The authors use the terms gustatory aversive learning, aversive learning, gustatory learning, and salt aversive learning interchangeably. They are not necessarily the same (esp. gustatory learning vs. aversive learning). Please pick a term and use it consistently.

Fig. 6C is so overlapped as to be impossible to read. Please either separate into several insets, or otherwise alter the figure so that the individual lines can be discerned.

It was not clear from the text what the difference is between *capa-1::TeTx::mCherry* #1 and #2. Please clarify, either in the figure legend or the results.

Testing the ASG calcium responses of *nmur-1* mutants (which could be reported in Fig. 4A, similar to Fig. 4B) would contribute additional evidence for the paper's core hypothesis. A null (no change from WT) result here would support the finding that *capa-1* / *nmur-1* signaling is activated during retrieval, not acquisition; i.e., the changes in ASG calcium response are due to a yet-unknown acquisition mechanism.

Reviewer #2:

Remarks to the Author:

Beets and colleagues present a very interesting manuscript describing the role of NMU peptide (CAPA-1) and the receptor NMUR-1 in salt avoidance behavior in *C. elegans*. The experimental details are well described and the data is presented beautifully. The statistical analyses appear appropriate for the conditions and variables tested.

The strengths of the manuscript are the multiple tests and different methodologies employed (ex. Txt and optogenetic silencing). The authors also have presented a very clear story from start to finish. Although I find the phenotypes presented to be of interest, I do question the overall conclusion made by the authors that this is a memory phenotype.

Are the terms learning and memory appropriate under the experimental paradigm used? Based in on the methods presented animals are exposed to a 15min training in high salt. How do the authors rule out the idea that the animals simply move to a quadrant that is more osmotically comfortable? The Strange lab has shown that *C. elegans* can adapt to 25-500mM NaCl, perhaps the 15min exposure to salt differentially effects the ability of the genetic mutants to maintain homeostasis or creates a new baseline set point? With the current data, it is unclear whether this is truly memory or just a point in the maintaining osmotic balance.

In a related point, the authors show that silencing CAPA-1 neurons during the acquisition phase had little effect on learned salt avoidance. This is an interesting finding, but is this a timing issue, based on relatively short exposure to the salt buffer?

I think this is an excellent study and I believe it will be of interest to the readership of Nature communications, but I believe the authors have not yet dissected a difference in physiological homeostasis (osmotic stress) from "learning".

Reviewer #3:

Remarks to the Author:

Watteyne and colleagues describe a novel neuromodulatory mechanism involved in salt avoidance. The major findings include the contribution of a neuropeptide, NMU to learned salt avoidance in *C. elegans*. The authors use a well-described paradigm in which animals are tested for their attraction to NaCl after association with a negative stimulus (no food). From these experiments, the authors find that the NMU receptor, *nmur-1* is required for gustatory learning after exposure to salt in the absence of food. Using a smart calcium mobilization assay *in vitro*, the authors find that *capa-1* neuropeptides are required for this learning, with NMUR-1 as their receptor. The authors show, using neuron-specific promoter experiments, that AFD neurons are likely the neuron expressing the receptor NMUR-1, as expression of *nmur-1* specifically in this cell type rescues the phenotypes of interest, which are almost entirely locomotory patterns: chemotaxis, klinotaxis, and biased random walk. Using transgenic techniques, the authors show that *capa-1* is expressed in the ASG neuron pair. They then use calcium imaging to show that ASG neurons change their responsivity after aversive conditioning. They also show that ASE (the salt sensor) is not altered by *nmur-1* mutations. Using optogenetic silencing (Arch) they show that the ASG neurons (*capa-1* expressing) are required during the 'retrieval' of the salt avoidance behavior, rather than the acquisition phase. They end with an interesting idea: CAPA-1 neurons (ASG) are not involved in innate salt attraction, but rather require a stressful or aversive situation in which the animal requires a heightened sensory paradigm.

Below are a list of larger issues or questions left unanswered by the manuscript in its current form. This list is followed by a shorter list of minor issues, including typographical errors.

Major issues:

- Behavioral assays appear to show a great deal of variability in sample sizes. For example, in Fig 2c the NaCl-conditioned traces of *nmur-1* are visibly more distinct from the wild-type trace than in Fig 2h. This makes the sudden increase in sample size in Fig 2h somewhat concerning, as significant changes were able to be seen in Figs 2c-f based on much fewer worms than in fig 2i. According to the caption, Fig 2c has $n > 25$ for each genotype, but it is unclear how many total times the assay was run if a single replicate of the assay requires a population of around 25 worms.
- The salt concentrations vary substantially between assays. In the quadrant assay, 100mM NaCl-conditioned worms are tested for their avoidance of 25mM NaCl. They also use a gradient assay, where worms are placed in the middle of a gradient to 100mM NaCl. Imaging is done with steps from 0mM NaCl to 50mM. These differences in NaCl concentration make it difficult to confidently connect data from different experiments with different setups.
- It is possible, since they cannot condition in their liquid-based assay high salt with the presence of food, that the lack of attraction to 25mM NaCl following 15 minutes of 100mM NaCl is habituation, not learning to avoid salt because it was paired with starvation. They address this issue in supplementary figure 1 by conditioning on agar plates which can be seeded with bacteria. They find that the absence of food during NaCl conditioning is necessary for aversive learning, and that *nmur-1* mutants are defective in the plate assay. However, they use an increased conditioning time (30 minutes) rather than the 15 minutes used in the liquid assay. Also, how likely is it that the worms are eating the food that they are exposed to? Finally, is *capa-1* also defective in the plate assay?
- In the abstract, the authors describe that the release of CAPA-1 neuropeptides are required for the learned salt avoidance, but they do not test for dependence on *unc-31*.
- The two navigational strategies, biased random walk and klinotaxis, are not well described. In Figure 2D, E and F, the readers would benefit from a description of the equation used to calculate these indices. Although they are somewhat described within the text, lines 153-155 are not clear. Also, sample size of figure 2h is much higher, making it easier to see significant differences.
- Would these findings be generalizable to other negative experiences? I suggest pairing salt with another stressor and testing the same learning paradigm.
- Raw numbers for these indices would also bolster the findings – fractional differences can often be misleading. The data could be shown as a supplementary table of these raw values.
- The authors make no attempt to describe why the learning paradigm only lasts for 10 minutes.

o Could overstimulation of AFD or ASG neurons (or overexpression of CAPA-1) increase this timeframe?

- In Figure 1, the diagram in C has a typo in the equation (the denominator should have a + sign rather than a – sign. Otherwise, your chemotactic index would always be 1.
- The authors never describe why the quadrant assay is used in the beginning, but the gradient assay is used later.
- The authors describe nmur-1/nmur-2 double mutants, but this work could really be strengthened by a double mutant of capa-1 and nmur-1. Is there a reason why this experiment was not performed?
- Is there evidence for the NaCl gradient truly being present from the angled plate pouring method?
- The data in Figure 6 follows animals for 10 minutes total and then describes the chemotactic, biased random walk, and klinotaxis indices for only 5 minutes, but no explanation is given for why they used different timings for these experiments.
- The authors summarize all of the statistics performed in the paper within a paragraph at the end of the Methods. It would aid in the reading and interpretation of the paper if there were at least references to the figures within this paragraph.

Minor:

- Figure 5A needs more labels in the legend. The dashed lines are not delineated within the three groups.
- Supplementary Figure 5 title capitalizes each word, while none of the other figure titles do.
- The authors are not consistent reporting statistical tests and number of experimental trials within the legends of figures.
- In Supplementary figure 8, the authors switch between HT115 and OP50 within a single sentence (Line 97-98).
- Figure 2, the legend skips the letter i.
- The labels for figures 3H and 3I are unclear.
- In lines 390 and 403, the lasers used to illuminate and silencing of ASG are described as “green-red” and “yellow-green” – what is the wavelength of light being used here?
- It is unclear how many animals are tested on each plate and whether the chemotactic, klinotactic, and biased random walk indices plot individual plates or averages from each day.
- Traces on figures are hard to distinguish from each other, especially in figure 6c
- Typos Figure 1 caption:
127 positive control⁴⁶. In this and all subsequent figures, data from mock-conditioned worms is depicted
128 in blue, while red markers denote NaCl-conditioned worms.
-

Response to Reviewer Comments

We thank all referees for their thoughtful and supportive reviews, and for their suggestions of ways to improve our manuscript. We have revised our work accordingly and added data from additional experiments including the cell-specific rescue of mutant phenotypes in population chemotaxis assays, the behavioral analysis of NMU signaling mutants in multiple paradigms testing gustatory aversive learning and osmotic homeostasis, the behavioral tracking of double mutants generated by CRISPR/Cas9, and the imaging of salt-evoked calcium responses in receptor-expressing AFD neurons.

We have addressed all reviewers' comments in a detailed response below. Changes to the manuscript text have been highlighted in blue.

Reviewer #1 (Remarks to the Author):

In this paper, Watteyne et al contend that CAPA-1 / NMUR-1 peptidergic signaling is necessary for memory retrieval in *C. elegans* gustatory aversive learning. The authors begin by identifying a gustatory aversive learning defect in mutants lacking *nmur-1*, and further localize the role of *nmur-1* to a single neuron pair (AFD). After identifying CAPA-1 as the neuropeptide that signals to NMUR-1, they demonstrate that inhibiting chemical signaling from the CAPA-1 expressing neuron pair (ASG) impairs gustatory aversive learning. Further, the authors silence the ASG neuron pair during either acquisition or retrieval; this demonstrates that ASG neuronal activity is necessary for retrieval, but not acquisition, of gustatory associative learning. The evidence presented is convincing (with one exception, noted as a major revision below), and the identification of a conserved neuropeptide as a modulator of memory retrieval is a valuable contribution to the study of learning and memory. I expect that the findings of this paper will be of interest to a wide readership.

We thank our reviewer for a fair and thoughtful review, and the interest in our work. We have endeavored to address her/his queries and critique below.

Major:

Overall, the data for the gene rescue experiments for *nmur-1* are poor. On the one hand, it is convincing that AFD is involved (given consistent results across three promoters). On the other hand, the variability of the WT control (which should be fairly stable) is at least as large, if not sometimes larger than, the genotype differences (e.g. compare WT chemotactic index between 3B and 3E, compared to the WT vs mutant chemotactic index in 3D). When compared to the initial WT and *nmur-1* mutant results (Fig. 2D-F), many of the “unsuccessful” rescues look more similar to WT than *nmur-1* mutants (e.g. supp. 2D, supp 2E). This creates some doubt regarding the gene rescue results.

Ideally, the authors should re-run these experiments using the original chemotaxis assay (Fig. 1D), which minimizes noise by using 100+ animals per assay. Additionally, using non-fluorescent mutant animals from co-injection experiments is a poor control. It allows for the possibility that some of the animals in the mutant group are “silent rescues” – worms with the rescue plasmid who did not get the co-injection fluorescence. I expect this may be the reason that, for instance, chemotactic index in Fig. 2I for *nmur-1* mutants is closer to the wildtype than the mutant in 2D, and that klinotaxis showed no difference between mutants and WTs in supp. 2A and supp. 3C. Ideally, the authors would repeat these experiments using an empty

plasmid with the co-injection marker for mutant controls; if that isn't feasible, using un-injected mutant controls would at least show an equivalent mutant phenotype for comparison.

The reviewer raises several concerns that we have thought carefully about; we address them by adding multiple experiments, which we detail throughout our responses below.

As suggested by the reviewer, we have repeated the experiments for cell-specific rescue of *nmur-1* phenotypes using the quadrant chemotaxis assay described in Fig. 1c. We find that expression of *nmur-1* under control of (i) the *nmur-1* promoter, (ii) the broad sensory neuron promoters *osm-6* and *ceh-23* (expressed in AFD amongst other neurons), and (iii) the AFD-specific *gcy-8* promoter all rescued the *nmur-1* phenotype (**Fig. 2b-c**). Promoters driving expression of *nmur-1* in sensory neurons other than AFD or in various interneurons did not rescue the mutant phenotype. These results (**Fig. 2b-c**) support our conclusions based on the rescue of salt chemotaxis behavior in the gradient assay (**Fig. 3f, Supplementary Fig. 4-5**), identifying AFD as a main site where *nmur-1* is required for gustatory aversive learning.

Prompted by our reviewer's question, we further investigated a role of AFD in salt chemotaxis behavior by performing ablation and calcium imaging experiments.

To test effects of AFD ablation, we used two transgenic strains expressing either the human caspase ICE or the split caspases TU#813 and TU#814 under the *gcy-8* promoter (Vidal-Gadea et al., 2015 *eLife*; Glauser et al., 2011 *Genetics*). Ablating AFD disrupted salt chemotaxis both in mock- and NaCl-conditioned animals (**Supplementary Fig. 6a-b**). In addition, AFD-ablated animals showed higher rates of immobility on the NaCl gradient (**Supplementary Fig. 6c**). We believe this salt chemotaxis defect does not emanate from non-specific ablation effects as both strains that we used in our assays have been shown to normally navigate in olfactory and noxious thermal environments (Vidal-Gadea et al., 2015 *eLife*; Glauser et al., 2011 *Genetics*).

In our calcium imaging experiments, we find that AFD responds to up- and downshifts in NaCl concentration, consistent with previous reports (Yemini et al., 2019 *bioRxiv*; Zaslaver et al., 2015 *PNAS*). We tested whether NaCl conditioning influences these responses but found only subtle effects on NaCl-evoked calcium activity in AFD (**Fig. 6b**). Mock- and NaCl-conditioned wild-type animals did not show significant differences in AFD calcium responses to up- and downshifts in NaCl. We also did not observe significant differences in AFD calcium responses of *nmur-1* mutants compared to wild type (**Fig. 6b**). These findings suggest that modulation of salt chemotaxis by *nmur-1* does not affect NaCl-evoked calcium activity at the cell body of AFD neurons.

The results detailed above further strengthen a role of AFD in the regulation of salt chemotaxis. We acknowledge, however, it cannot be ruled out that NMUR-1 signaling in additional neurons contributes to gustatory aversive learning. We note this in the discussion in lines 412-415.

We apologize that our description of controls used in rescue experiments was unclear, as we did not use non-fluorescent mutant animals as controls. All rescue experiments were done using un-injected mutant animals as controls. We have clarified this in the methods sections in lines 750-755.

Several experiments (e.g. many of the mock groups in Fig. 1D, supp. 5A-B) have very small N – it looks like as low as 3. That is really too few. Off the cuff, I would suggest a minimum of 10. N should be added to any experiments with low N, and the authors should either be

relatively conservative in making sure to include enough N, or include a power analysis to provide a basis for their decision to use low N. Additionally, it is not clear why there are sometimes drastic differences in the numbers of animals between experiments (e.g. Fig. 2D-F use “at least 25 animals”, while the same assay in Fig. 2I uses “at least 284 animals”). This should be addressed.

We agree and have addressed this issue by increasing N for all experiments with low N numbers. To this end, we have performed additional quadrant chemotaxis assays, aiming at a minimum N of 10 assays for each condition and/or genotype (**Fig. 1d, Supplementary Fig. 1**).

We chose to have a higher N-number for behavioral tracking experiments because, compared to the quadrant assay, this assay is more sensitive to variation due to the fact that individual worms are measured rather than populations. We have re-run experiments and analysis of tracking data using N numbers in the order of 50 to 100 animals for each condition and/or genotype (**Fig. 3b-f, Supplementary Fig. 4-5**).

Minor:

Fig. 1C has a typo – the denominator should be $n(A) + n(C)$, not -.

This has been corrected.

It would be nice to have a representative picture of the worms in the salt gradient assay (to allow the reader to assess the imaging setup – contrast, resolution, etc).

We have included a representative figure of a worm on the NaCl gradient in **Supplementary Fig. 2a**.

Several experiments use custom code created in the lab. It would be nice if that code would be packaged as part of the supplemental, to allow readers to check the code independently.

We understand the reviewers’ request. However, we prefer to provide our custom code upon request and have noted this in the Methods section in lines 601 and 684-686. Our motivation for this decision is that the current form of our code is not sufficiently user-friendly, as it still requires manual quality checks that have to be communicated to potential users and also have to be tailored to other experimental setups and parameters.

The authors should clarify how outliers were determined, and whether they were included in statistical analysis between groups.

We always included outliers in the statistical analyses. Outliers were automatically flagged by the standard MATLAB `boxplot()` function. They are depicted in figure panels as data points scattered beyond the whiskers of the boxplots. We have clarified and added this information in the legend of Fig. 1 and in the Methods section in lines 718-721.

The authors use the terms gustatory aversive learning, aversive learning, gustatory learning, and salt aversive learning interchangeably. They are not necessarily the same (esp. gustatory learning vs. aversive learning). Please pick a term and use it consistently.

We agree and have now consistently used gustatory aversive learning throughout the revised manuscript.

Fig. 6C is so overlapped as to be impossible to read. Please either separate into several insets, or otherwise alter the figure so that the individual lines can be discerned.

We have adapted this figure, now **Fig. 7g** in the revised manuscript, to include average instead of individual traces.

It was not clear from the text what the difference is between *capa-1::TeTx::mCherry* #1 and #2. Please clarify, either in the figure legend or the results.

The *capa-1::TeTx::mCherry* #1 and #2 labels indicated two independent strains expressing tetanus toxin under the *capa-1* promoter. Both strains showed significant and similar defects in gustatory aversive learning. To avoid confusion, we have included data of only one strain in **Fig. 7a-d** and stated that the results were confirmed with a second independent strain in the legend of this figure.

Testing the ASG calcium responses of *nmur-1* mutants (which could be reported in Fig. 4A, similar to Fig. 4B) would contribute additional evidence for the paper's core hypothesis. A null (no change from WT) result here would support the finding that *capa-1* / *nmur-1* signaling is activated during retrieval, not acquisition; i.e., the changes in ASG calcium response are due to a yet-unknown acquisition mechanism.

We believe that ASG calcium imaging in our setup cannot provide clear temporal information on the activity of *capa-1* / *nmur-1* signaling. ASG calcium responses reported in Fig. 4a, now **Fig. 6a** of the revised manuscript, were tested after mock- or NaCl-conditioning of animals, i.e. during retrieval. If we were to observe similar calcium responses in *nmur-1* mutant animals, we can only conclude that the experience-dependent modulation of ASG calcium activity does not require *nmur-1*. This result, however, would not provide evidence that *capa-1* / *nmur-1* signaling is activated specifically during retrieval. NMUR-1 signaling may affect other neuronal signaling mechanisms than NaCl-evoked calcium activity, as suggested by AFD calcium imaging (see above and Fig. 6b). If *nmur-1* mutants were to show different ASG calcium responses compared to wild type, this result would also not reject a role of *nmur-1* signaling specifically during retrieval. One possible scenario is that feedback from *nmur-1* neurons (e.g. AFD) to ASG during retrieval contributes to alterations in ASG calcium responses and in salt chemotaxis behavior.

Reviewer #2 (Remarks to the Author):

Beets and colleagues present a very interesting manuscript describing the role of NMU peptide (CAPA-1) and the receptor NMUR-1 in salt avoidance behavior in *C. elegans*. The experimental details are well described and the data is presented beautifully. The statistical analyses appear appropriate for the conditions and variables tested.

The strengths of the manuscript are the multiple tests and different methodologies employed (ex. Txt and optogenetic silencing). The authors also have presented a very clear story from start to finish. Although I find the phenotypes presented to be of interest, I do question the overall conclusion made by the authors that this is a memory phenotype.

We thank the reviewer for his/her support of our work. We have now added data from additional experiments to further assess the learning and memory phenotypes, as detailed in our responses below.

Are the terms learning and memory appropriate under the experimental paradigm used? Based on the methods presented animals are exposed to a 15min training in high salt. How do the authors rule out the idea that the animals simply move to a quadrant that is more osmotically comfortable? The Strange lab has shown that *C. elegans* can adapt to 25-500mM NaCl, perhaps the 15min exposure to salt differentially effects the ability of the genetic mutants to maintain homeostasis or creates a new baseline set point? With the current data, it is unclear whether this is truly memory or just a point in the maintaining osmotic balance.

This is an important point that we have addressed by performing two additional experiments:

Experiment 1: We first asked if the switch in NaCl chemotaxis behavior could be a non-specific effect of the exposure to high osmolarity during conditioning with salt. Therefore, we replaced NaCl in our conditioning protocol with glycerol at the same osmolarity and tested the effect on NaCl chemotaxis behavior. Animals were exposed for 15 min to 200 mM glycerol (the same osmolarity as 100 mM NaCl) in the absence of food, after which chemotaxis to 25 mM NaCl was assayed on quadrant plates. We find that pre-exposing wild-type worms to glycerol did not affect chemotaxis to NaCl and did not result in salt avoidance (**Supplementary Fig. 1b**), which is in accordance with previous reports (Saeki et al., 2001 *J Exp Biol*). By contrast, wild-type worms pre-exposed to 100 mM NaCl in the absence of food did show salt aversion. This result indicates that the conditioning effect of NaCl on salt chemotaxis behavior cannot be attributed to its osmolarity.

Our data for *capa-1* and *nmur-1* animals suggest that these mutants show normal behavioral responses to osmotic stimuli. In the initial version of our manuscript we showed that *capa-1* and *nmur-1* mutants display normal chemotaxis behavior at different NaCl concentrations (100 to 400 mM NaCl, **Supplementary Fig. 1a**). In addition, *capa-1* and *nmur-1* animals had no defects in NaCl chemotaxis behavior after conditioning with glycerol (**Supplementary Fig. 1b**). This suggests that salt chemotaxis behavior is not differentially affected by exposure to increased osmolarity in the NMU signaling mutants.

Experiment 2: We also asked the reverse question if *capa-1* and *nmur-1* animals differentially respond to increased osmolarity after conditioning with NaCl. To test this, we replaced 25 mM NaCl in our chemotaxis quadrant plates with glycerol at the same osmolarity (50 mM) and tested chemotaxis behavior of mock- and NaCl-conditioned animals (**Supplementary Fig. 1c**). Under these conditions, wild-type and mutant animals showed little attraction or aversion to glycerol. For each genotype the chemotactic indices of mock- and NaCl-conditioned animals were close to zero, although the values after NaCl conditioning were significantly lower than those of the mock condition. This result suggests that NaCl conditioning affects the chemotaxis response to osmotic stimuli other than NaCl. However, this effect is much smaller than the switch in chemotaxis behavior to NaCl itself. This finding confirms that the latter is primarily a learned response rather than the result of acclimation to an altered osmotic environment. Importantly, mock- and NaCl-conditioned *capa-1* and *nmur-1* animals showed normal chemotaxis to glycerol compared to wild type (**Supplementary Fig. 1c**). By contrast, mutants of *capa-1* and *nmur-1* had clear defects in salt chemotaxis behavior after conditioning with NaCl (**Fig. 1d and 5a**). These results indicate that loss of

capa-1 and *nmur-1* function disrupts gustatory aversive learning rather than the ability to maintain osmotic balance.

In a related point, the authors show that silencing CAPA-1 neurons during the acquisition phase had little effect on learned salt avoidance. This is an interesting finding, but is this a timing issue, based on relatively short exposure to the salt buffer?

We thank the reviewer for this remark. Prompted by this question, we asked whether impaired *capa-1* / *nmur-1* signaling disrupts learned salt avoidance after longer conditioning times. We performed a time course experiment in which wild-type and *nmur-1* mutant animals were mock- or NaCl-conditioned for 0, 7.5, 15, 30 and 60 minutes (**Supplementary Fig. 2d-e**). Mutants of *nmur-1* were still impaired in gustatory aversive learning compared to wild type with increased conditioning time. This suggests that the learning phenotype of *nmur-1* animals is not due to a change in the conditioning time required for learning.

A recent report of the Iino lab (Jang et al., 2019 *PNAS*) supports our findings that ASG neurons are required for the expression of learned NaCl aversion, although this work did not identify molecular players in ASG involved in learning. Jang and colleagues used the histamine-gated chloride channel HisCl to silence ASG by addition of histamine during either NaCl starvation conditioning (lasting 6 hours) or subsequent salt chemotaxis behavior (lasting 30 minutes). NaCl-conditioned behavior was only statistically different between histamine-treated and untreated animals when ASG was silenced during NaCl chemotaxis behavior, not during conditioning. Thus, even long-term silencing of ASG during the 6-hour conditioning period in their protocol did not result in a significant reduction of learned salt avoidance. This finding and our results suggest that ASG is required for the expression of learned salt aversion. We have included this reference in the discussion.

I think this is an excellent study and I believe it will be of interest to the readership of Nature communications, but I believe the authors have not yet dissected a difference in physiological homeostasis (osmotic stress) from "learning".

We thank the reviewer for his/her interest in our manuscript and refer to our response above discussing two additional experiments that we performed to confirm the role of *capa-1* / *nmur-1* signaling in learning.

Reviewer #3 (Remarks to the Author):

Watteyne and colleagues describe a novel neuromodulatory mechanism involved in salt avoidance. The major findings include the contribution of a neuropeptide, NMU to learned salt avoidance in *C. elegans*. The authors use a well-described paradigm in which animals are tested for their attraction to NaCl after association with a negative stimulus (no food). From these experiments, the authors find that the NMU receptor, *nmur-1* is required for gustatory learning after exposure to salt in the absence of food. Using a smart calcium mobilization assay in vitro, the authors find that *capa-1* neuropeptides are required for this learning, with NMUR-1 as their receptor. The authors show, using neuron-specific promoter experiments, that AFD neurons are likely the neuron expressing the receptor NMUR-1, as expression of *nmur-1* specifically in this cell type rescues the phenotypes of interest, which are almost entirely locomotory patterns: chemotaxis, klinotaxis, and biased random walk. Using transgenic techniques, the authors show that *capa-1* is expressed in the ASG neuron pair. They then use calcium imaging to show that ASG neurons change their responsivity after

aversive conditioning. They also show that ASE (the salt sensor) is not altered by *nmur-1* mutations. Using optogenetic silencing (Arch) they show that the ASG neurons (*capa-1* expressing) are required during the ‘retrieval’ of the salt avoidance behavior, rather than the acquisition phase. They end with an interesting idea: CAPA-1 neurons (ASG) are not involved in innate salt attraction, but rather require a stressful or aversive situation in which the animal requires a heightened sensory paradigm.

Below are a list of larger issues or questions left unanswered by the manuscript in its current form. This list is followed by a shorter list of minor issues, including typographical errors.

Behavioral assays appear to show a great deal of variability in sample sizes. For example, in Fig 2c the NaCl-conditioned traces of *nmur-1* are visibly more distinct from the wild-type trace than in Fig 2h. This makes the sudden increase in sample size in Fig 2h somewhat concerning, as significant changes were able to be seen in Figs 2c-f based on much fewer worms than in fig 2i. According to the caption, Fig 2c has $n > 25$ for each genotype, but it is unclear how many total times the assay was run if a single replicate of the assay requires a population of around 25 worms.

Please see also our answer to comment 2 of reviewer 1. Briefly:

- We have increased N for all quadrant chemotaxis assays with low N numbers to ensure that all experiments have a consistent number of replicates, aiming at a minimum N of 10 assays for each condition (**Fig. 1d, Fig. 2b-c, Fig. 5a, Supplementary Fig. 1**).

- For behavioral tracking experiments we chose to have a higher N-number, because this assay tracks individual worms rather than populations and is therefore more sensitive to variation. We have re-run our experiments and analysis of tracking data using N numbers in the range of 50 to 100 animals for each condition in behavioral tracking assays (**Fig. 3b-f, Fig. 5b-f, Supplementary Fig. 3-8**).

- We have included information on the number of experimental repeats in the Methods section in lines 738-750.

The salt concentrations vary substantially between assays. In the quadrant assay, 100mM NaCl-conditioned worms are tested for their avoidance of 25mM NaCl. They also use a gradient assay, where worms are placed in the middle of a gradient to 100mM NaCl. Imaging is done with steps from 0mM NaCl to 50mM. These differences in NaCl concentration make it difficult to confidently connect data from different experiments with different setups.

We have restructured the results section (lines 143-153 and 299-303) and we have added data comparing chemotaxis on different salt gradients for better clarification of the rationale for the NaCl concentrations used.

Our initial focus was on gustatory aversive learning that we first tested in an established quadrant assay following published experimental procedures (Hukema et al., 2006 *EMBO J.*, 2008 *Learn Mem*; Beets et al., 2012 *Science*). This assay provides a robust end-point measurement of learned salt aversion that is amenable to testing different pre-exposure conditions, but only quantifies the distribution of worms on the quadrants. Previous work has shown that learned salt aversion depends on the experience-dependent modulation of locomotory patterns, i.e. klinotaxis and biased random walk, using NaCl gradients (Luo et al.,

2014 *Neuron*; Kunitomo et al., 2013 *Nat Comm*; Iino and Yoshida et al., 2009 *J. Neurosci.*). Thus, having shown that *nmur-1* is required for learned salt aversion in the quadrant assay (**Fig. 1d**), it made sense to investigate the effect of deleting *nmur-1* on klinotaxis and biased random walk. As expected, *nmur-1* mutants were defective in the experience-dependent modulation of these locomotory parameters on a linear NaCl gradient (**Fig. 3b-e**).

Although different NaCl gradients were used for quadrant assays and tracking of locomotory parameters, the phenotypes of *nmur-1* and *capa-1* mutants were consistent between these assays (lines 171-202 and 221-234). Firstly, in both assays *nmur-1* and *capa-1* mutants showed defects in learned salt aversion (**Fig. 1d, 3b-f, 5a-c, Supplementary Fig. 7a**). Secondly, additional cell-specific rescue experiments (see reviewer 1) yield similar findings on the cellular focus of *nmur-1*, required in AFD, in both the quadrant and behavioral tracking assay (**Fig. 2b-c and Fig. 3f**). Thirdly, in response to this reviewer's comment, we also compared *nmur-1* phenotypes on different linear salt gradients. We tracked salt chemotaxis behavior of mock- and NaCl-conditioned animals on linear gradients of 0 to 50 mM, 0 to 100 mM and 0 to 150 mM NaCl (**Supplementary Fig. 3**). We find that *nmur-1* mutants are defective in learned salt aversion on all gradients tested. Thus, mutant phenotypes and rescue data are consistent across different chemotaxis assays and NaCl gradients, which provides solid evidence for *capa-1* / *nmur-1* functions in learned salt aversion.

For calcium imaging, we measured responses to a 0 to 50 mM NaCl upshift as well as a 50 to 0 mM NaCl downshift for two reasons (**Fig. 6**): Firstly, at the start of our tracking experiments worms are put at the center of a linear NaCl gradient (0 to 100 mM), where they experience approximately 50 mM NaCl. Thus, imaging calcium responses upon shifts to/from 50 mM NaCl is relevant to the behavioral data. Secondly, published calcium imaging studies have used similar concentration shifts for the neurons we imaged (f.i. Zaslaver et al., 2015 *PNAS* for AFD imaging; Murayama et al., 2013 *Curr Biol* for ASG imaging), allowing the comparison of our findings to previous work.

It is possible, since they cannot condition in their liquid-based assay high salt with the presence of food, that the lack of attraction to 25mM NaCl following 15 minutes of 100mM NaCl is habituation, not learning to avoid salt because it was paired with starvation. They address this issue in supplementary figure 1 by conditioning on agar plates which can be seeded with bacteria. They find that the absence of food during NaCl conditioning is necessary for aversive learning, and that *nmur-1* mutants are defective in the plate assay. However, they use an increased conditioning time (30 minutes) rather than the 15 minutes used in the liquid assay. Also, how likely is it that the worms are eating the food that they are exposed to? Finally, is *capa-1* is also defective in the plate assay?

Our reviewer is right about the increased conditioning time in the plate assays and we have now pointed out this difference in the results section (lines 111-115). The increased conditioning time for assays on plate was based on Hukema et al. (2008, *Learn Mem*), who reported that conditioning for 30 minutes is required for robust learning behavior in this assay. During this time window, worms in all likelihood will eat the food they are exposed to, because the *C. elegans* pharynx exerts on average 250 pumps/minute when feeding on *E. coli* OP50 bacteria (Lee et al., 2015 *Sci Rep*; Kudelska et al., 2018 *Invert Neurosci*). To our knowledge, no reports have documented the suppression of feeding behaviors in the presence of 100 mM NaCl, which is considered an attractive stimulus rather than a noxious cue (Hukema et al., 2006 *EMBO J*). In *C. elegans*, hypertonic stress resulting in salt avoidance

only occurs when NaCl concentrations exceed 200 mM (Hukema et al., 2006 *EMBO J*). Throughout the animal kingdom salty flavors indicate the presence of nutritious compounds and generally lead to ingestion (Geerling and Loewy 2008 *Exp Physiol*; Hurley and Johnson 2015 *Pflugers Arch*). We therefore believe that the concentration of 100 mM NaCl does not influence basal feeding rates and that sensitivity to this NaCl concentration is not different between wild type and *capa-1 / nmur-1* mutants based on our work in Supplementary Fig. 1a. We also investigated whether conditioning *nmur-1* for 30 minutes in liquid would result in the same salt aversive learning phenotype as compared to conditioning for only 15 minutes, which we found was indeed the case (**Supplementary Fig. 2d-e**). Together with our data from assays using conditioning on plates, this shows that *nmur-1* animals are defective in associating salt with the absence of food.

For *capa-1* mutants we now also provide data for learned salt aversion after conditioning on plates (**Supplementary Fig. 1d**) or in liquid (**Fig. 5a**). In both assays *capa-1* mutant animals are indeed defective in gustatory aversive learning.

In the abstract, the authors describe that the release of CAPA-1 neuropeptides are required for the learned salt avoidance, but they do not test for dependence on *unc-31*.

We thank the reviewer for this suggestion. We have now tested the dependence on *unc-31* using the *ets-5p::unc-31* RNAi strain from Juozaityte et al. (*PNAS*, 2017), silencing *unc-31* in ASG and BAG neurons. As expected, this strain was defective in learned salt aversion and showed behavioral defects similar to those observed for knockdown of *capa-1* (**Fig. 5f and Supplementary Fig. 7e**). ASG-selective knockdown of both *capa-1* and *unc-31* did not result in an additive defect, suggesting that *unc-31* and *capa-1* function in the same genetic pathway.

The two navigational strategies, biased random walk and klinotaxis, are not well described. In Figure 2D, E and F, the readers would benefit from a description of the equation used to calculate these indices. Although they are somewhat described within the text, lines 153-155 are not clear. Also, sample size of figure 2h is much higher, making it is easier to see significant differences.

To clarify quantifications of the different navigational strategies, we restructured Fig. 3 and added descriptions of the equations used to calculate all indices on the figure panels and legends of **Fig. 3c-e**. We also expanded our description of indices in the Methods section (lines 620-641).

Our reviewer is right about the sample size of Fig. 2H and this has now been addressed in the revised manuscript (**Supplementary Fig. 5d**) as detailed in the response to our reviewer's first comment.

Would these findings be generalizable to other negative experiences? I suggest pairing salt with another stressor and testing the same learning paradigm.

This is an interesting question that we have addressed, as the reviewer suggested, by pairing salt with another stressor than the absence of food. Previous work has shown that pairing NaCl with aversive concentrations of the odorant benzaldehyde results in learned salt aversion (Hukema et al., 2008 *Learn Mem*). Using this training protocol, we tested whether *nmur-1* and *capa-1* mutants are defective in salt aversion after training with NaCl and

benzaldehyde. Indeed, we find that *nmur-1* and *capa-1* mutants displayed learning defects in this paradigm, suggesting that aversive learning driven by negative experiences other than the absence of food are also governed by NMU signaling (**Supplementary Fig. 1e**).

Raw numbers for these indices would also bolster the findings – fractional differences can often be misleading. The data could be shown as a supplementary table of these raw values.

The reviewer is correct in remarking that behavioral indices are computed as a fractional measure for chemotactic behavior of individual animals in tracking experiments. We have, however, handled several quality constraints to ensure that the fractional differences faithfully reflect the behaviors under study. First, as detailed in the Methods section (lines 628-629), only individual trajectories of sufficient duration and quality are used for behavioral analysis. This way, we minimized noise introduced by short tracks (caused by inter-animal collisions, harsh pre-treatment or tracking issues). Second, some experimental conditions might display overall defects in locomotion, which would non-specifically result in defects in the behavioral metrics for chemotaxis. This is not the case as worms defective for NMU signaling still display normal locomotion behaviors on and off food (**Supplementary Fig. 9**). For conditions in which this was the case, such as when tracking NaCl-conditioned behavior of animals in which AFD is genetically ablated (**Supplementary Fig. 6**), we chose to quantify the fraction of time animals were immobile in addition to the mean position and the individual chemotactic indices, which are all measures not disproportionately affected by locomotory defects. Because of these quality controls and the large amount of tracking data, we did not provide raw numbers of indices in a Supplementary Table format, but can share this data upon request as noted in the manuscript (lines 1035-1038).

The authors make no attempt to describe why the learning paradigm only lasts for 10 minutes.

Previous work showed that gustatory aversive learning induced by 15 min of conditioning lasts for 10-15 min when assayed on salt quadrant plates and reported that 10 min of chemotaxis is sufficient to test for learning defects (Jansen et al., 2002 *EMBO J.*, Hukema et al., 2006 *EMBO J.*, 2008 *Learn Mem*). Therefore, we have consistently calculated chemotaxis indices after 10 minutes when testing NaCl chemotaxis behavior on quadrant plates.

In chemotaxis assays with linear NaCl gradients, animals were tracked for 30 minutes on the gradient after conditioning. We find that wild-type animals, when mock-conditioned, migrate up the gradient immediately after the start of the assay and overall maintain this position on the gradient over subsequent time windows (**Fig. 3b and Supplementary Fig. 2c-d**). Similarly, NaCl-conditioned animals migrate to lower mean positions on the NaCl gradient in the first ten minutes and display gradual displacement back to higher salt concentrations over time. Thus, the chemotactic drive is strongest in the first 10-min time interval, after which the chemotactic index for both mock- and NaCl-conditioned worms averages zero in the 10 to 20 and 20 to 30 min time intervals (**Supplementary Fig. 2g**). In tracking experiments we therefore calculated behavioral indices in the 0 to 10 min time window, similar to the duration of the quadrant assay. We find that the magnitude of the aversive response to NaCl depends on the duration of training with salt in the absence food (see reviewer 2 – **Supplementary Fig. 2d and 2f**).

We have further clarified the duration of the learning paradigms in the results section, lines 82-85 and lines 158-165.

Could overstimulation of AFD or ASG neurons (or overexpression of CAPA-1) increase this timeframe?

We thank our reviewer for this interesting question. We first tried an optogenetic approach to test if overstimulation of AFD or ASG, using ChR2 or bPAC, influences learned salt aversion. These experiments require assays to be performed in a *lite-1* mutant background in which the natural photophobic response to blue light, used for activation of ChR2 and bPAC, is disrupted. The *lite-1 (ce314)* allele is most commonly used as a genetic background for optogenetic experiments (Edwards et al., 2008 *PLoS Biol* and Husson et al., 2013 *Biol Cell*). However, we find that *lite-1 (ce314)* mutants have defects in NaCl chemotaxis behavior (see data below) and therefore, we cannot use this genetic background in our behavioral assays.

Following the reviewer's suggestion, we then turned to overexpression of *capa-1* and examined the effect on learned salt aversion. We find that overexpression of *capa-1* under its own promoter did not affect NaCl chemotaxis behavior of mock- and NaCl-conditioned animals (**Supplementary Fig. 7c**). Here again, the mean position and behavioral indices were quantified in the first 10 minutes after release on the NaCl gradient, which is the time interval in which the magnitude of the conditioned response is a measure for the learned NaCl aversion (see comment above). We obtained similar results for three independent overexpression strains. This finding suggests that the effects of CAPA-1 signaling on learned salt aversion may not depend on peptide expression levels but rather on the regulation of timing for their release from ASG, which is also supported by our results from optogenetic silencing of ASG.

In Figure 1, the diagram in C has a typo in the equation (the denominator should have a + sign rather than a - sign. Otherwise, your chemotactic index would always be 1.

This has been corrected.

The authors never describe why the quadrant assay is used in the beginning, but the gradient assay is used later.

We have restructured the results (lines 143-153) in order to clarify the rationale for why we expand our behavioral analysis to include locomotory patterns using the gradient assay in the latter part of the manuscript.

Our initial focus was on gustatory aversive learning that we first tested in an established quadrant assay following published experimental procedures (Hukema et al., 2006 *EMBO J.*, 2008 *Learn Mem*; Beets et al., 2012 *Science*). This assay provides a robust end-point measurement of learned salt aversion that is amenable to testing different pre-exposure conditions. Previous work, using linear NaCl gradients, has shown that learned salt aversion depends on the experience-dependent modulation of locomotory patterns, i.e. klinotaxis and biased random walk (Luo et al., 2014 *Neuron*). Having shown that *nmur-1* is required for learned salt aversion in the quadrant assay, it made sense to investigate the effect of deleting *nmur-1* on klinotaxis and biased random walk. As expected, mutants of *nmur-1* were defective in the experience-dependent modulation of both locomotory parameters on linear NaCl gradients (**Fig. 3b-e**). Phenotypes and rescue data of *capa-1* and *nmur-1* mutants were consistent between the quadrant and gradient assays (**Fig. 1d, 2b-c, 3b-f, 5a-c. Supplementary Fig. 2-5**), supporting a role of NMUR-1 signaling in learned salt aversion. Because the gradient assay provides a more detailed quantification of locomotory patterns involved in gustatory aversive learning, we chose to use this assay in all subsequent experiments.

The authors describe *nmur-1/nmur-2* double mutants, but this work could really be strengthened by a double mutant of *capa-1* and *nmur-1*. Is there a reason why this experiment was not performed?

We agree with the reviewer that assaying a *capa-1; nmur-1* double mutant could strengthen the conclusions of our manuscript. We previously attempted to generate this strain through genetic crossing, but failed repeatedly as both genes are closely linked on the X chromosome. We have now used a CRISPR strategy to create a deletion allele of *nmur-1*, similar to the *nmur-1(ok1387)* deletion, in both the wild-type and the *capa-1(ok3065)* mutant background. As expected, the *capa-1(ok3065); nmur-1(lst1672)* double mutant recapitulated the learning defect of the *capa-1(ok3065)* and *nmur-1(lst1672)* single mutants, which supports that *capa-1* and *nmur-1* function in the same genetic pathway (**Fig. 5b-c and Supplementary Fig. 7b**).

Is there evidence for the NaCl gradient truly being present from the angled plate pouring method?

The method used for generating linear NaCl gradients is based on Luo et al. 2014 *Neuron*. This study included a numerical simulation of the NaCl diffusion as it generates the stable linear gradient. We have verified the linear relationship of NaCl concentrations along the y-axis of the square plates using a chloride electrode for three different NaCl gradients (0 to 50 mM NaCl, 0 to 100 mM NaCl and 0 to 150 mM NaCl). This analysis showed a linear increase in chloride concentrations (**Supplementary Fig. 2b**), confirming that the NaCl gradient is indeed formed.

The data in Figure 6 follows animals for 10 minutes total and then describes the chemotactic, biased random walk, and klinotaxis indices for only 5 minutes, but no explanation is given for why they used different timings for these experiments.

The reviewer is right about the time windows used for optogenetic experiments and we have clarified the rationale for different timings in the results (lines 346-348 and Figure 7 legend in lines 1178-1179). For ASG silencing, we had to adapt assay plates with linear NaCl gradients to fit our microscope setup for optogenetic stimulation and tracking. The 12 x 12 cm square plates used in all population behavioral tracking experiments were too large to fit the

motorized microscope stage required to keep individual animals in the middle of the light beam for optogenetic silencing during retrieval. Therefore, we used circular Petri dishes of 55 mm diameter as assay plates in this setup. Because salt gradients of 0 to 100 mM NaCl are technically challenging to establish in these smaller plates, we used linear salt gradients of 0 to 50 mM NaCl and reduced the time windows for tracking and analysis to accommodate these shorter gradients.

The authors summarize all of the statistics performed in the paper within a paragraph at the end of the Methods. It would aid in the reading and interpretation of the paper if there were at least references to the figures within this paragraph.

We apologize for this confusion and have now consistently included a description of the statistical analysis in each figure legend and have elaborated the explanation of statistical analyses in the Methods section (lines 724-737).

Minor:

Figure 5A needs more labels in the legend. The dashed lines are not delineated within the three groups.

We thank the reviewer for pointing out this mistake and have corrected the legend of this figure panel (**Fig. 7a** in the revised manuscript).

Supplementary Figure 5 title capitalizes each word, while none of the other figure titles do.

This has been corrected. The data from Supplementary Figure 5 has now been added to **Supplementary Fig. 1a and 1d**.

The authors are not consistent reporting statistical tests and number of experimental trials within the legends of figures.

This is an important point and we have now consistently reported statistical tests and experimental trials within the figure legends and in the Methods section.

In Supplementary figure 8, the authors switch between HT115 and OP50 within a single sentence (Line 97-98).

This has been corrected. We thank the reviewer for pointing out this error.

Figure 2, the legend skips the letter i.

We apologize for this mistake. Legends in Fig. 2 (Fig. 3 in the revised manuscript) have been adapted.

The labels for figures 3H and 3I are unclear.

The labels in both figures have been adjusted. For the *capa-1* rescue panel (**Fig. 5e**), we have indicated that a *capa-1* rescue transgene is expressed in a *capa-1* mutant background. For the ASG-selective *unc-31* and *capa-1* RNAi experiments (**Fig. 5f**), we have specified each RNAi

transgene and denoted its presence in each of the conditions tested in a wild-type genetic background.

In lines 390 and 403, the lasers used to illuminate and silencing of ASG are described as “green-red” and “yellow-green” – what is the wavelength of light being used here?

We have referred to our Methods section where the used wavelengths for optogenetics are specified (lines 342-343 and lines 710-715). For ASG silencing during conditioning, animals were kept under yellow-green light (540 – 580 nm) on a Leica MZ16F microscope fitted with an EL6000 external light source and mCherry Leica filter set (#10450195). For silencing during retrieval, animals were continuously illuminated with green-red light (567 – 602 nm) during tracking on a Zeiss AxioObserver Z1 microscope fitted with a Colibri.2 LED module and Filter cube 61 HE GFP/HcRed.

It is unclear how many animals are tested on each plate and whether the chemotactic, klinotactic, and biased random walk indices plot individual plates or averages from each day.

We have specified this in the Methods section (lines 738-749). Briefly, 20 to 30 animals were put at the center of the linear NaCl gradient at the start of the experiment. Individual migration trajectories (or tracks) were then delineated from video recordings by the tracking algorithm discussed in the Methods section. Tracks of poor quality (f.i. tracks shorter than 5 minutes within the respective indicated time interval – commonly caused by collisions with the arena or other animals) were discarded before analysis. Chemotactic, biased random walk and klinotaxis indices were calculated for each remaining individual track within the experiment. For all behavioral assays, data was collected on at least two independent days, which typically pooled 6 – 10 experimental repeats. This resulted in animal numbers allowing robust statistical analysis (sample sizes for tracking experiments are the number of worms).

Traces on figures are hard to distinguish from each other, especially in figure 6c.

To clarify data in this figure panel (**Fig. 7g** of revised manuscript) we have now plotted average traces rather than individual traces.

Typos Figure 1 caption: positive control. In this and all subsequent figures, data from mock-conditioned worms is depicted in blue, while red markers denote NaCl-conditioned worms.

We have corrected this typo, and again thank all our reviewers for their thoughtful comments.

Reviewers' Comments:

Reviewer #1:

Remarks to the Author:

The authors have made satisfactory revisions to all of my initial concerns, as well as included several additional follow-up experiments that enhance the potential impact of the paper. I recommend accepting the paper without further revisions.

Reviewer #2:

Remarks to the Author:

The authors have addressed my concerns. Moreover they have provided a detailed and rich set of new data to address the concerns raised by the other reviewers.

Reviewer #3:

Remarks to the Author:

With their revised version of the paper, the authors addressed several concerns and add some interesting new data. They improve their evidence that *nmur-1* is important in AFD neurons by showing multiple promoter rescues. The addition of gustatory aversive learning in response to benzaldehyde was especially interesting. They also image the relevant neurons, AFD and ASG, in response to salt. However, some concerns remain. The writing is somewhat convoluted in that it is not clear what question they are trying to answer with each experiment and what the data from each experiment means for their model. For example, they show that ASG responds differently to NaCl-off compared after training but do not comment on how this fits into their model. While they added extra explanations for their rationale behind the setup of the experiments, they stop at the level of using precedent set in previous literature. Their explanations instead should be specifically tailored to the points that they are trying to prove to present a cohesive story. The result is that the paper appears to lack intention.

Response to Reviewer Comments

We thank all referees for their thoughtful and supportive comments on our revised manuscript. Following the suggestions of reviewer 3, we have rewritten the introduction to clarify the rationale and intention of our study, emphasizing previous work that led us to hypothesize a role for NMU signaling in associative learning and the rationale to investigate the function of NMU signaling in *C. elegans* neurobiology and behavior.

We have addressed all reviewers' comments in a detailed response below.

REVIEWERS' COMMENTS:

Reviewer #1 (Remarks to the Author):

The authors have made satisfactory revisions to all of my initial concerns, as well as included several additional follow-up experiments that enhance the potential impact of the paper. I recommend accepting the paper without further revisions.

We thank the reviewer for the helpful comments that greatly improved the manuscript.

Reviewer #2 (Remarks to the Author):

The authors have addressed my concerns. Moreover they have provided a detailed and rich set of new data to address the concerns raised by the other reviewers.

We wish to thank reviewer #2 for the positive feedback and suggestions that have allowed us to contextualize our findings further.

Reviewer #3 (Remarks to the Author):

With their revised version of the paper, the authors addressed several concerns and add some interesting new data. They improve their evidence that *nmur-1* is important in AFD neurons by showing multiple promoter rescues. The addition of gustatory aversive learning in response to benzaldehyde was especially interesting. They also image the relevant neurons, AFD and ASG, in response to salt. However, some concerns remain. The writing is somewhat convoluted in that it is not clear what question they are trying to answer with each experiment and what the data from each experiment means for their model. For example, they show that ASG responds differently to NaCl-off compared after training but do not comment on how this fits into their model. While they added extra explanations for their rationale behind the setup of the experiments, they stop at the level of using precedent set in previous literature. Their explanations instead should be specifically tailored to the points that they are trying to prove to present a cohesive story. The result is that the paper appears to lack intention.

We thank the reviewer for the great suggestions for our experimental work and interpretation of the data. Following this latest comment, we have substantially re-framed the introduction as to clearly convey:

- the role of neuromodulators in shaping neural circuit function underlying learning and memory

- the extent to which the understudied class of neuropeptide neuromodulators is expected to participate in experience-dependent behavior
- previous work that suggest NMU signaling to have an evolutionarily conserved role in the modulation of learning circuits
- the current knowledge of NMU signaling in *C. elegans*, and why we hypothesize this evolutionarily conserved signaling system influences gustatory aversive learning

Furthermore, we have restructured the discussion as to logically fit our experimental findings into the conceptual model, and further contextualize this study in the broader context of neuropeptidergic signaling in learning and memory:

- CAPA-1/NMUR-1 signaling as general regulator of gustatory aversive learning
- a defined temporal role of CAPA-1 neuropeptides within learned salt aversion
- aversive conditioning alters NaCl-evoked responses in the gustatory circuit, including the primary salt sensor ASE and *capa-1*-expressing ASG neurons, which is hypothesized to contribute to learned salt aversion
- CAPA-1 peptides signal through the NMUR-1 receptor on the AFD sensory neurons, which participate in the generation of salt chemotaxis behavior
- NMU signaling is conserved across bilaterian animals, and could therefore play similar cognitive functions in other animals

We hope our revised manuscript now presents a more cohesive story that is clearly and specifically introduced and discussed.